# MEDCCO: UNLEASHING OPEN-ENDED REASONING IN MEDICAL MULTI-MODAL LANGUAGE MODELS VIA CURRICULUM REINFORCEMENT LEARNING

## ABSTRACT

Recent advances in reinforcement learning with verifiable, rule-based rewards have greatly enhanced the reasoning capabilities and out-of-distribution generalization of VLMs/LLMs, obviating the need for manually crafted reasoning chains. Despite these promising developments in the general domain, their translation to medical imaging remains limited. Besides, current reinforcement fine-tuning (RFT) approaches in medical reasoning are primarily designed for close-ended visual question answering (VQA), where answer choices are provided within the query. This narrow focus limits the model's capacity to leverage world knowledge and adapt to diverse clinical tasks. More importantly, such methods fail to meet the pressing clinical need for open-ended, reasoning-intensive decision-making, which requires generating answers without predefined options—a task proven much more challenging. To bridge this gap, we propose **MedCCO**, the first multi-modal reinforcement learning framework for medical VQA that integrates both close-ended and open-ended data under a curriculum-based RFT strategy. By explicitly fostering open-ended reasoning, MedCCO aims to enhance performance across both reasoning types. Specifically, MedCCO is initially fine-tuned on a diverse set of close-ended medical VQA tasks to establish domain-grounded reasoning capabilities, and is then progressively adapted to open-ended tasks to foster deeper knowledge enhancement and clinical interpretability. We validate MedCCO across eight challenging medical VQA benchmarks, spanning both close-ended and open-ended settings. Experimental results show that MedCCO consistently enhances performance and generalization, achieving a 11.4% accuracy gain across three in-domain tasks, and a 5.7% improvement on five out-of-domain benchmarks. These findings highlight the promise of curriculum-guided RL in advancing robust, clinically-relevant reasoning in medical multi-modal language models.[1]

## 1 INTRODUCTION

Medical vision-language models (VLMs) have demonstrated significant advancements (Moor et al., 2023; Wu et al., 2023; Huang et al., 2025) in lesion detection and clinical diagnosis, driven primarily by supervised fine-tuning (SFT) on large-scale annotated datasets. Nonetheless, medical imaging tasks inherently demand more than mere accuracy in visual interpretation; they require transparent, clinically relevant rationales underpinning each diagnostic decision. Furthermore, the capacity for open-ended reasoning is frequently essential in real-world clinical settings, playing a critical role in trustworthy decision-making and timely therapeutic interventions.

Recent studies (DeepSeek-AI, 2025; Team, 2025; Du et al., 2025; Wang et al., 2025; Huang et al., 2025) indicate that reinforcement learning (RL) is highly effective at enhancing sophisticated reasoning capabilities in large language models (LLMs) and general VLMs, obviating the necessity for meticulously crafted, high-quality long chain-of-thought (CoT (Wei et al., 2022)) annotations. However, these promising results have yet to be thoroughly replicated or systematically validated within medical VLM contexts. Moreover, current RL-based methodologies applied to medical VLMs (Lai et al., 2025; Pan et al., 2025) primarily target close-ended VQA tasks, often limited to perception-oriented assessments that probe basic visual comprehension (Zuo et al., 2025), e.g., modality and body parts. Such a restricted focus substantially constrains the models' ability to cultivate deeper, open-ended reasoning capabilities that are critical in clinical applications.

To tackle this issue, we propose **MedCCO**, the first multi-modal reasoning framework that employs a curriculum-driven reinforcement learning paradigm for both medical close- and open-ended VQA tasks. Our objective

---

[1]Code and model weights are available at: https://anonymous.4open.science/r/MedCCO/.

is to unleash the open-ended reasoning within the reinforcement learning framework to enhance the model's performance on close-ended reasoning, while simultaneously improving its capability in free-text reasoning to better address real-world clinical demands. For open-ended RFT, we introduce a hybrid reward function that incorporates both lexical similarity metrics (BLEU and ROUGE) and semantic equivalence measures (BERTScore) to jointly promote surface-level fluency and deep semantic coherence. We explore a couple of joint Group Relative Policy Optimization (GRPO) based RL training strategies and find that sequential knowledge injection through a curriculum-based approach yields superior results. Specifically, **MedCCO** is trained in a two-stage manner: it initially acquires domain-specific reasoning capabilities via close-ended medical VQA under reinforcement learning, and is subsequently adapted to more challenging open-ended tasks in a progressive fashion. This joint training scheme promotes both discriminative accuracy and generative flexibility, enabling the model to retrieve world knowledge and construct structured reasoning chains without relying on hand-crafted annotations. We conducted extensive experiments to assess the universality of **MedCCO** for medical VQA tasks, performing a systematic comparison across both close-ended and open-ended tasks. **MedCCO** consistently surpasses competitive baselines on in-domain and out-of-domain test sets, and evaluations on the SLAKE (Liu et al., 2021) benchmark further demonstrate its cross-modal robustness.

Additionally, an ablation study on the joint RL paradigm revealed that a curriculum approach, first training on close-ended examples, then progressively introducing open-ended ones, outperforms training on both task types simultaneously. We attribute this gain to the conflicts between discrete and continuous reward signals and the differing difficulty levels of the two tasks. Together, these results suggest a promising reinforcement-learning paradigm in which new reasoning capabilities can be injected to model sequentially, without sacrificing previously acquired performance.

To summarize, our contributions are four-fold:

1. To the best of our knowledge, we propose the first multi-modal medical reasoning model capable of handling both close-ended and open-ended VQA tasks within a unified framework;
2. We introduce a novel hybrid reward function that integrates lexical and semantic metrics to jointly optimize textual fluency and semantic coherence, which proves effective for open-ended RFT.
3. We explore a curriculum reinforcement fine-tuning strategy that enables the VLM to learn from simple to complex tasks while retaining previously acquired knowledge.
4. We conduct extensive experiments on both in-domain and out-of-domain medical VQA datasets. **MedCCO** achieves state-of-the-art performance and, on several benchmarks, matches or surpasses the performance of larger models significantly.

## 2 RELATED WORK

**General and Medical VLMs.** Alignment-based methods such as CLIP (Radford et al., 2021) and BLIP-2 (Li et al., 2023c) paved the way for vision-language models. Subsequent work (Liu et al., 2023; Chen et al., 2024b; Bai et al., 2025; Li et al., 2023a) demonstrates that instruction tuning with a few hundred thousand high-quality image–text pairs can unlock strong VQA and multi-modal reasoning capabilities. This paradigm has been extended to the medical domain by models like Med-Flamingo (Moor et al., 2023), LLaVA-Med (Li et al., 2023b), and HuatuoGPT-Vision (Chen et al., 2024a), which combine large-scale medical image–text alignment with domain-specific supervision to support accurate visual understanding and basic clinical diagnosis.

**Reinforcement learning in LLMs/VLMs.** Reinforcement learning has been widely adopted to align VLMs and LLMs with human preferences and mitigate hallucinations (Sun et al., 2023; Yu et al., 2024a;b; Zhao et al., 2023; Zhou et al., 2024). GRPO-based (DeepSeek-AI, 2025) methods have demonstrated strong effectiveness in enhancing the reasoning capabilities of VLMs/LLMs through rule-based rewards and got broadly verified in general visual reasoning scenarios (Huang et al., 2025; Liu et al., 2025; Zhou et al.; Zhang et al., 2025; Shen et al., 2025). However, such progress in the general domain has yet to be systematically validated in medical imaging. In this work, we extend this line of research to medical VQA by introducing a novel RL framework that uniquely integrates both close-ended and open-ended VQA data, and conduct comprehensive empirical evaluations to demonstrate its effectiveness.

**Reinforcement Learning in Medical LLMs/VLMs.** Recent studies have begun exploring reinforcement learning (RL) as a scalable alternative to supervised fine-tuning for enhancing medical reasoning in LLMs and VLMs. In LLMs, FineMedLM-o1 (Yu et al., 2025) and MedReason (Wu et al., 2025) expand the training paradigm via test-time adaptation and structured knowledge supervision, while HuatuoGPT-o1 (Chen et al., 2024a) leverages verifiable problems to improve reasoning through RL. In contrast, progress in VLMs remains limited. Recent

efforts such as Med-R1 (Lai et al., 2025) and MedVLM-R1 (Pan et al., 2025) show that rule-based RL can enhance generalization in medical VQA, but focus solely on close-ended questions, neglecting unified reasoning across modalities. To address this gap, our work applies GRPO-based RL within a unified framework encompassing both close-ended and open-ended VQA. To support this, we design task-specific rewards and adopt a curriculum training strategy that progressively enhances the model's reasoning capabilities.

## 3 METHODOLOGY

Medical VQA tasks can be broadly categorized into close-ended and open-ended formats. As shown in Figure 1 (b), close-ended VQA involves selecting from predefined options or verifying statements, while open-ended VQA requires free-form generation, demanding deeper reasoning over medical knowledge. In this work, we propose **MedCCO**, an extension of vanilla GRPO that integrates both types of VQA to enhance reasoning capabilities in medical VLMs. Following the curriculum learning strategy illustrated in Figure 1 (a), MedCCO is first trained on close-ended tasks with accuracy-based rewards, and subsequently fine-tuned on open-ended tasks, which are inherently more challenging. To stabilize reinforcement learning on open-ended data, we enhance VQA data consistency using an advanced VLM. By leveraging open-ended supervision, MedCCO improves not only close-ended performance but also fosters more autonomous and flexible reasoning.

## (a) MedCCO: curriculum close and open ended GRPO

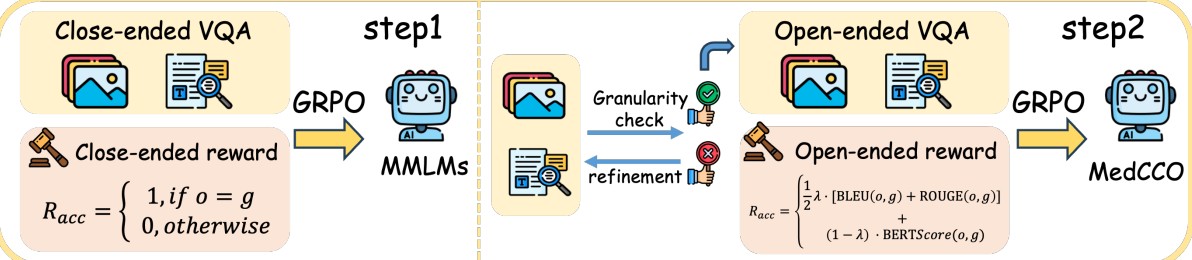

## (b) Applications of MedCCO

| close-ended reasoning | open-ended reasoning |
|---|---|

**close-ended reasoning:** : What is the diagnosis? (A) Echinococcal infection (B) Pneumatosis intestinalis (C) Trichobezoar (D) Meckel's diverticulitis (E) Adrenal cancer

: <think> The image appears to be a CT scan of the abdomen. The presence of air within the bowel loops is characteristic of pneumatosis intestinalis, which is a condition where air accumulates in the walls of the intestines. This can be seen as gas-filled structures within the bowel wall on a CT scan. There is no evidence of an echinococcal infection, trichobezoar, or Meckel's diverticulitis that would typically present with different imaging findings. Adrenal cancer is less likely to be diagnosed through a CT scan of the abdomen alone without additional imaging of the adrenal glands. </think> <answer> B </answer>

**open-ended reasoning:** : Identify the primary anatomical system depicted in the image.

: <think> The image appears to be a histological section of tissue, likely from a glandular organ given the presence of glandular structures with a stratified epithelium. The architecture suggests it could be from the gastrointestinal tract, possibly the stomach or small intestine, due to the characteristic folds and glands. The red coloration is indicative of a staining technique such as hematoxylin and eosin (H&E), which is commonly used in histology to highlight different components of the tissue. The presence of connective tissue and blood vessels further supports this interpretation. </think> <answer> Gastrointestinal </answer>

Figure 1: Overview of our MedCCO. MedCCO is first fine-tuned with GRPO on close-ended medical VQA to establish fundamental reasoning capabilities, then adapted to open-ended tasks to enable more advanced reasoning and knowledge retrieval. Prior to open-ended GRPO, we refine VQA data consistency to improve training efficiency.

Table 1: Prompt template used in MedCCO. The placeholder {Question} is replaced with specific reasoning questions during training.

---

You are a helpful assistant. {Question} Output the thinking process in `<think> </think>` and final answer in `<answer> </answer>` tags. The output answer format should be as follows: `<think>` reasoning process here `</think><answer>` answer here (Do not provide any explanation) `</answer>` Please strictly follow the format.

---

### 3.1 BACKBONE LEARNING FRAMEWORK

We build upon recent works (Lai et al., 2025; Pan et al., 2025) that apply Group Relative Policy Optimization (GRPO) to improve the reasoning abilities of medical VLMs using close-ended VQA data. GRPO (Shao et al., 2024) is a reinforcement learning algorithm similar to PPO (Schulman et al., 2017), with two key differences: (1) GRPO operates in a value-free regime by computing generalized advantage estimation (GAE) using group-relative rewards; and (2) it employs verifiable rule-based outcomes as rewards instead of relying on pre-trained reward models. At each training step, the model generates $G$ candidate responses $\{o_i\}_{i=1}^{G}$ from the current policy $\pi_{\theta_{\text{old}}}$. Each output is assigned a scalar reward $r_i$ based on rule-based criteria. The advantage is normalized by the mean and standard deviation of all group rewards:

$$A_i = \frac{r_i - \text{mean}(\{r_j\}_{j=1}^{G})}{\text{std}(\{r_j\}_{j=1}^{G})}. \tag{1}$$

The GRPO objective is defined as:

$$\mathcal{L}_{\text{GRPO}}(\theta) = -\frac{1}{\sum_{i=1}^{G}|o_i|}\sum_{i=1}^{G}\sum_{t=1}^{|o_i|}\left[\min\left(\frac{\pi_\theta(o_{i,t} \mid q, o_{i,<t})}{\pi_{\theta_{\text{old}}}(o_{i,t} \mid q, o_{i,<t})}\hat{A}_{i,t},\right.\right.$$
$$\left.\left.\text{clip}\left(\frac{\pi_\theta(o_{i,t} \mid q, o_{i,<t})}{\pi_{\theta_{\text{old}}}(o_{i,t} \mid q, o_{i,<t})}, 1-\epsilon, 1+\epsilon\right)\hat{A}_{i,t}\right) - \beta\mathbb{D}_{\text{KL}}[\pi_\theta\|\pi_{\text{ref}}]\right], \tag{2}$$

where $\mathbb{D}_{\text{KL}}(\pi_\theta\|\pi_{\text{ref}})$ serves as a regularization term to penalize divergence from the reference policy $\pi_{\text{ref}}$, with $\beta$ controlling its strength.

### 3.2 MULTI-REWARD POLICY

To effectively guide reinforcement learning and prevent reward hacking, we design a multi-dimensional reward schema targeting three aspects: correctness, semantic alignment, and format adherence. Each component targets a distinct behavioral dimension. During training, models are prompted using the instruction shown in Table 1, and structured responses are parsed to compute corresponding rewards.

**Close-ended Reward.** For close-ended tasks, we use a binary reward to enforce strict correctness:

$$R_{\text{close}}(o, g) = \begin{cases} 1, & \text{if } o = g, \\ 0, & \text{otherwise,} \end{cases} \tag{3}$$

where $o$ is the predicted answer and $g$ is the ground truth.

**Open-ended Rewards.** For open-ended responses, we introduce a hybrid reward function that jointly captures surface-level fidelity and deep semantic alignment. Given that the ground-truth answers are relatively short (with a mean length of approximately 5 tokens), we employ BLEU-1 and ROUGE-1 to assess lexical overlap, while BERTScore is used to evaluate semantic similarity.

$$R_{\text{open}}(o, g) = \frac{1}{2}\lambda \cdot (\text{BLEU}_1(o, g) + \text{ROUGE}_1(o, g)) + (1 - \lambda) \cdot \text{BERTScore}(o, g), \tag{4}$$

where $\lambda \in [0, 1]$ controls the trade-off between lexical and semantic metrics.

**Format Reward.** To ensure structured output, we impose a format reward (denoted as $R_{\text{format}}$) that checks for compliance with required tags. Specifically, the reasoning content must be enclosed in `<think> ⋯ </think>` and the answer in `<answer> ⋯ </answer>` tags respectively.

**Total Reward.** For each sample, the total reward is computed as $0.8R + 0.2R_{\text{format}}$, where $R$ denotes $R_{\text{close}}$ for close-ended and $R_{\text{open}}$ for open-ended questions.

### 3.3 JOINT REINFORCEMENT LEARNING TRAINING STRATEGIES

To combine close-ended and open-ended VQA tasks under a unified reinforcement learning framework, we explore two gradient-based reinforcement policy optimization strategies based on GRPO.

**Direct Joint GRPO via Gradient Re-weighting.** We first investigate joint GRPO by simultaneously optimizing over both task types. Due to the inherently different reward structures: discrete for close-ended and continuous for open-ended, the resulting reward advantages exhibit distinct variance characteristics, which in turn affect the gradient magnitudes during optimization. To address this, we re-weight the gradients of each task within the same mini-batch to balance their contributions. Specifically, given a mini-batch $\mathcal{B} = \mathcal{B}_c \cup \mathcal{B}_o$, where $\mathcal{B}_c$ and $\mathcal{B}_o$ denote close- and open-ended samples respectively, we compute per-sample gradients $g^{(k)} = \nabla_\theta \mathcal{L}_{\text{GRPO}}(\theta)$, and task-wise averaged gradients are:

$$\bar{g}_c = \frac{1}{|\mathcal{B}_c|} \sum_{i \in \mathcal{B}_c} g^{(i)}, \quad \bar{g}_o = \frac{1}{|\mathcal{B}_o|} \sum_{j \in \mathcal{B}_o} g^{(j)}, \tag{5}$$

and the final combined gradient is:

$$\mathcal{G} = \alpha \cdot \bar{g}_c + (1 - \alpha) \cdot \bar{g}_o, \quad \alpha = \frac{|\mathcal{B}_o|}{|\mathcal{B}_c| + |\mathcal{B}_o|}. \tag{6}$$

where the mixing coefficient $\alpha \in (0, 1)$ is statically set based on the relative batch sizes of close-ended and open-ended samples, ensuring proportionally balanced gradient contributions across tasks.

**Curriculum GRPO.** We further explore a curriculum-based strategy to address the challenges of applying RL to open-ended VQA. Unlike close-ended tasks with well-defined answer spaces, open-ended VQA requires the VLM to generate free-form responses that align closely with ground truth—an inherently difficult task in the medical domain where domain knowledge is essential. This often results in unstable and inefficient learning. Inspired by curriculum learning (Bengio et al., 2009), we adopt a progressive training scheme to gradually build reasoning capability. As illustrated in Figure 1, the model is first trained on close-ended questions using GRPO to establish a stable policy, followed by fine-tuning on open-ended data. Empirically, this curriculum-driven GRPO approach stabilizes learning and consistently improves reasoning performance across both question types.

### 3.4 VQA DATA QUALITY REFINEMENT

Prior to using open-ended VQA data for GRPO, as depicted in Figure 1 (a), we perform a VQA consistency check and refinement. A major issue observed in current open-ended medical VQA datasets is the inconsistency in granularity between questions and answers, which impairs the learning process. While supervised fine-tuning primarily maps input-output patterns, reinforcement learning requires the model to align its reasoning with semantic details. For instance, in SLAKE (Liu et al., 2021) dataset, the question *"How was the image taken?"* may yield answers like *"CT"* or *"axial"*, which differ in specificity and introduce ambiguity. To address this, we implement a VQA-Consistency Auditor (i.e. Qwen2.5-VL (Bai et al., 2025)-72B) that refines noisy VQA pairs based on three core principles: (1) **Consistency evaluation:** ensuring the question comprehensively captures the semantic content of the answer; (2) **Open-ended enforcement:** maintaining free-form phrasing to support descriptive outputs; and (3) **Granularity matching:** aligning the specificity of the question with the answer to reduce under- or over-generalization. Detailed criteria and prompts for this refinement process are included in Appendix A.6.

## 4 EXPERIMENTS

**Training Datasets.** We utilize three publicly available medical VQA datasets—VQA-RAD (Lau et al., 2018), SLAKE (Liu et al., 2021) (English part), and PathVQA (He et al., 2020) as in-domain training datasets. Together, these datasets provide a total of 14,379 close-ended and 12,996 open-ended question-answer pairs, comprising approximately 27k training samples. For in-domain evaluation, we adhere to the official data splits provided by each dataset. To evaluate cross-modal generalization, we leverage the SLAKE dataset (Liu et al., 2021), which spans three imaging modalities: X-ray, MRI, and CT, each comprising both open-ended and close-ended samples. We train the model on a single modality and evaluate it across others' test sets. More details are in the Appendix.

**Benchmarks.** We adopt the same experimental settings and benchmarks as in (Chen et al., 2024a), with additional focus on recent reasoning datasets (enhanced coverage and difficulty), i.e., test sets of VQA-RAD (Lau et al., 2018), SLAKE (Liu et al., 2021), PathVQA (He et al., 2020), PMC-VQA (Zhang et al., 2023), and the Health & Medicine track of MMMU (Yue et al., 2024) for multi-modality handling. We include Quilt-VQA (Seyfioglu et al., 2024)

Table 2: Performance of our MedCCO, Medical VLMs and General VLMs on three in-domain datasets and three out-of-domain datasets. The best and second-best results in each column are highlighted in red and blue, respectively. Avg.: mean results of open-ended metrics. c.: close-end accuracy. o.: open-ended metrics (combined BLEU1, ROUGE1 and BERTScore, $\lambda = 0.7$.). $\phi$: LLM-as-judge evaluation. R: reasoning, U: understanding.

| Model | In-domain test | | | | | | | | | | Out-of-domain test | | | | | | |
|---|---|---|---|---|---|---|---|---|---|---|---|---|---|---|---|---|---|
| | VQA-RAD | | | SLAKE | | | Path-VQA | | | Avg. | Quilt-VQA | | | PMC | MedXpert | | Avg |
| | c. | o. | $\phi$ | c. | o. | $\phi$ | c. | o. | $\phi$ | | c. | o. | $\phi$ | c. | R.c. | U.c. | |
| *General Zero-shot VLMs* | | | | | | | | | | | | | | | | | |
| Yi-VL-34B | 53.0 | 22.4 | 66.2 | 58.9 | 33.9 | 65.1 | 47.3 | 12.9 | 73.8 | 38.1 | 56.0 | 13.2 | 53.2 | 39.5 | 19.9 | 20.7 | 29.9 |
| LLaVA-v1.6-7B | 52.6 | 19.8 | 65.3 | 57.9 | 37.6 | 68.4 | 47.9 | 12.6 | 78.0 | 38.1 | 58.3 | 8.7 | 55.7 | 35.5 | 20.7 | 20.6 | 28.8 |
| LLaVA-v1.6-13B | 55.8 | 24.0 | 68.4 | 58.9 | 44.5 | 71.9 | 51.9 | 12.8 | 70.5 | 41.3 | 57.4 | 24.5 | 60.6 | 36.6 | 19.5 | 18.1 | 31.2 |
| LLaVA-v1.6-34B | 58.6 | 24.1 | 75.1 | 67.3 | 44.6 | 72.5 | 59.1 | 15.0 | 76.4 | 44.8 | 62.4 | 23.7 | 60.8 | 44.4 | 20.6 | 25.5 | 35.3 |
| Qwen2.5-VL-7B | 67.3 | 32.2 | 76.5 | 71.6 | 40.2 | 70.4 | 65.5 | 17.2 | 78.5 | 49.0 | 54.8 | 29.0 | 62.9 | 50.4 | 20.6 | 23.1 | 35.6 |
| *Medical Zero-shot VLMs* | | | | | | | | | | | | | | | | | |
| Med-Flamingo-7B | 45.4 | 29.3 | 74.8 | 43.5 | 30.1 | 63.0 | 54.7 | 28.7 | 83.4 | 38.6 | 62.1 | 22.3 | 60.7 | 23.3 | 19.0 | 20.0 | 29.3 |
| RadFM-13B | 50.6 | 34.0 | 73.3 | 34.6 | 44.2 | 75.0 | 38.7 | 19.9 | 82.5 | 37.0 | 60.7 | 21.5 | 59.2 | 25.9 | 19.8 | 19.6 | 29.5 |
| LLaVA-Med-7B | 51.4 | 10.1 | 75.1 | 48.6 | 6.6 | 69.7 | 56.8 | 8.4 | 85.6 | 30.3 | 63.0 | 29.3 | 65.5 | 24.7 | 20.5 | 19.5 | 31.4 |
| HuatuoGPT-Vision-8B | 63.8 | 36.0 | 83.0 | 74.5 | 47.0 | 83.5 | 59.9 | 23.2 | 86.8 | 50.7 | 63.9 | 38.5 | 66.8 | 52.7 | 20.4 | 22.9 | 39.6 |
| *Finetuned VLM* | | | | | | | | | | | | | | | | | |
| Qwen2.5-VL-7B (SFT) | 71.3 | 27.8 | 76.5 | 78.6 | 50.8 | 79.8 | 87.8 | 33.6 | 80.2 | 58.3 | 60.9 | 8.9 | 64.5 | 49.2 | 20.2 | 20.4 | 31.9 |
| Qwen2.5-VL-7B (GRPO) | 70.5 | 29.8 | 77.8 | 79.3 | 40.2 | 76.8 | 82.8 | 27.8 | 81.4 | 55.1 | 50.2 | 28.4 | 66.6 | 51.2 | 21.2 | 21.7 | 34.5 |
| Joint-GRPO-7B | 72.1 | - | - | 75.7 | - | - | 84.7 | - | - | - | 67.2 | - | - | 53.6 | 22.1 | 22.9 | - |
| MedCCO-7B | 76.3 | 40.0 | 84.9 | 79.4 | 65.7 | 85.6 | 82.8 | 29.6 | 88.9 | 62.3 | 69.4 | 39.3 | 70.5 | 53.2 | 23.2 | 23.6 | 41.7 |

for out-of-domain evaluation on open-ended questions, and MedXpertQA (Zuo et al., 2025)(MM part), a recently proposed benchmark emphasizing complex medical reasoning.

**Baselines.** Following (Lai et al., 2025), we compare our MedCCO's performance with Gerneral Zero-shot VLMs: Yi-VL (Young et al., 2024), LLaVA-v1.6 (Liu et al., 2024) and Qwen2.5-VL (Bai et al., 2025); Medical Zero-shot VLMs: Med-Flamingo (Moor et al., 2023), RadFM (Wu et al., 2023), LLaVA-Med (Li et al., 2023b), and HuatuoGPT-Vision (Chen et al., 2024a). There is limited research on applying reinforcement learning to enhance reasoning in medical VLMs. Recent efforts such as MedR1 (Lai et al., 2025) and MedVLM-R1 (Pan et al., 2025) employed vanilla GRPO for different medical tasks. Thus, we also compare our approach with vanilla GRPO fine-tuning, denoted as Qwen2.5-VL (GRPO), trained on close-ended data.

**Implementation Details.** Training is conducted using $4\times$H100 GPUs (80GB VRAM each) with PyTorch, leveraging FlashAttention-2 for computational efficiency. GRPO-related experiments are implemented using the `verl` (Sheng et al., 2024) framework to accelerate the training. We adopt the Qwen2.5-VL (Bai et al., 2025)-7B-Instruct model and its 3B variant as the backbone models. The training uses a total batch size of 64, with a learning rate of $1e^{-6}$. The KL penalty coefficient is set to $\beta = 0.01$, with $\lambda = 0.7$. For inference during GRPO, we deploy the model using `vllm` (Kwon et al., 2023) on 2 GPUs, generating 10 completions per sample, corresponding to the group size $G$ in GRPO. The maximum number of input pixels is capped at 401,408, and the temperature for `vllm` sampling is set to 1.0. GRPO training is conducted for 1 epoch and about 8 hours. For supervised fine-tuning (SFT), all experiments are performed using `LLaMAFactory` (Zheng et al., 2024), with its default configuration.

### 4.1 COMPARISON RESULTS

As shown in Table 2, MedCCO sets a new state-of-the-art across a wide range of medical VQA benchmarks, including both in-domain and out-of-domain evaluations. It achieves an average accuracy of 62.3% on in-domain datasets and 41.7% on out-of-domain ones, outperforming existing generalist and domain-specific baselines on 8 out of 11 sub-tasks. Compared to the second-best model, HuatuoGPT-Vision (Chen et al., 2024a)-8B, MedCCO achieves a 11.6% gain in in-domain accuracy while maintaining stronger generalization to novel tasks such as MedXpertQA (Zuo et al., 2025) and Quilt-VQA (Seyfioglu et al., 2024). In open-ended evaluation, MedCCO achieves state-of-the-art performance on three out of four benchmarks. We further perform an **LLM-as-judge evaluation in Appendix A.4**, and the results confirm the consistent superiority of MedCCO. Collectively, these

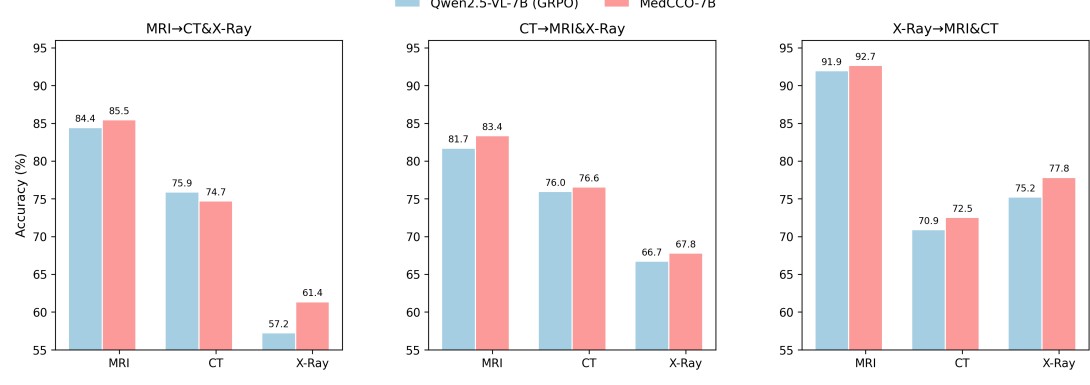

Figure 2: Cross-modal performance on SLAKE (Liu et al., 2021), with each model trained on a single modality and evaluated across all modalities for in- and cross-modal comparison.

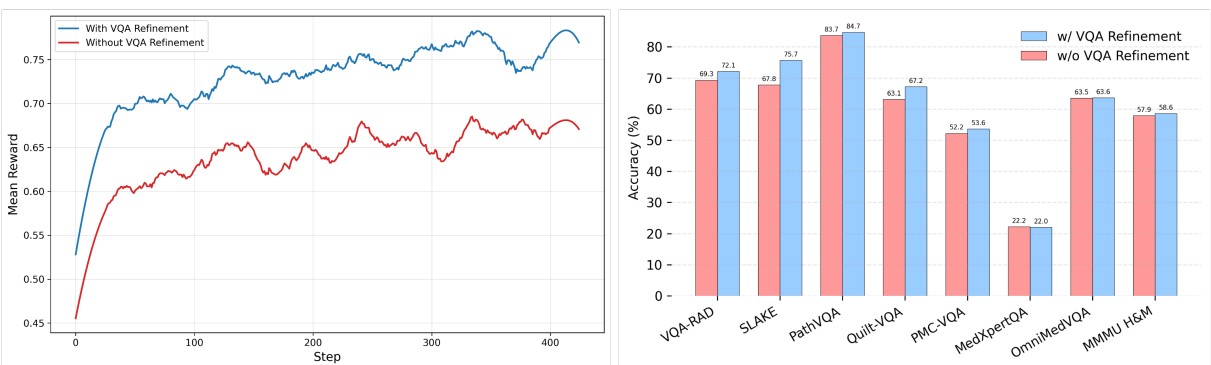

Figure 3: Training curves and overall performance with and without VQA refinement. Incorporating VQA refinement leads to more stable training and improved generalization.

findings demonstrate that MedCCO enhances both close-ended and open-ended reasoning abilities, underscoring the efficacy of our unified curriculum-guided GRPO approach in improving medical reasoning across various settings.

As shown in Table 3, MedCCO achieves the highest overall accuracy of 59.3% on the MMMU benchmark, surpassing all general zero-shot, medical-specific, and fine-tuned VLMs. Notably, it outperforms HuatuoGPT-Vision (Huang et al., 2025)-8B by over 10.2% in absolute accuracy, despite the latter being trained on more than 1M image–text pairs. These results underscore MedCCO's strength in comprehensive multi-modal understanding and its capacity for integrating visual and medical knowledge effectively.

To assess cross-modal transferability, we conduct experiments on the SLAKE (Liu et al., 2021) dataset by training on a single modality and evaluating on all three (MRI, CT, X-ray). Test sets refer to the union of the validation and test splits from the official SLAKE (Liu et al., 2021) dataset, which differs from the configuration used in Table 2. As shown in Figure 2, MedCCO-7B consistently outperforms Qwen2.5-VL (Bai et al., 2025)-7B using GRPO at all transfer scenarios. Most notably, under the most challenging X-ray → MRI & CT setting, MedCCO attains an accuracy of 92.7% on MRI and 72.5% on CT, achieving the best results across all modalities.

**Qualitative Analysis.** Figure 4 showcases representative examples of MedCCO's reasoning capability. In the open-ended cases (a) and (b), the model demonstrates accurate anatomical identification and physiological interpretation. In (a), it recognizes the lungs in a chest X-ray and correctly infers their primary function as breathe. In (b), the model identifies two kidneys in a cross-sectional CT scan based on their spatial location and morphological features. For the close-ended example (c), MedCCO handles a clinically grounded diagnostic task by selecting surgical excision as the next management step for a shoulder mass, justifying its decision with multi-modal evidence, including MRI findings and absence of metastasis on CT. These examples highlight the model's ability to integrate visual cues with medical knowledge to generate both descriptive and decision-oriented responses. For additional qualitative examples, please refer to Appendix A.7.

Table 3: Generalization (zero-shot) comparison on the selected MMMU (Yue et al., 2024) Health & Medicine track with category-wise and overall accuracy. The best and second-best results in each column are highlighted in red and blue, respectively. BMS: Basic Medical Science, CM: Clinical Medicine, DLM: Diagnostics and Laboratory Medicine, P: Pharmacy, PH: Public Health.

| Model | BMS | CM | DLM | P | PH | MMMU
Health & Medicine |
|---|---|---|---|---|---|---|
| *General VLM* | | | | | | |
| Yi-VL (Young et al., 2024)-34B | 49.4 | 48.9 | 43.2 | 40.5 | 32.0 | 41.5 |
| LLaVA-v1.6 (Liu et al., 2024)-7B | 40.5 | 36.9 | 32.1 | 32.3 | 26.9 | 33.1 |
| LLaVA-v1.6 (Liu et al., 2024)-13B | 53.6 | 46.7 | 33.3 | 22.2 | 40.0 | 39.3 |
| LLaVA-v1.6 (Liu et al., 2024)-34B | 56.4 | 56.0 | 46.9 | 46.7 | 41.7 | 48.8 |
| LLaVA-v1.5-LLaMA3-8B | 42.3 | 44.0 | 37.0 | 34.7 | 35.2 | 38.2 |
| Qwen2.5-VL (Bai et al., 2025)-7B | 50.0 | 63.3 | 33.3 | 59.3 | 53.3 | 51.7 |
| *Medical VLM* | | | | | | |
| Med-Flamingo (Moor et al., 2023) | 29.6 | 28.1 | 24.8 | 25.3 | 31.2 | 28.3 |
| RadFM (Wu et al., 2023) | 27.5 | 26.8 | 25.8 | 24.7 | 29.1 | 27.0 |
| LLaVA-Med (Li et al., 2023b)-7B | 39.9 | 39.1 | 34.6 | 37.4 | 34.0 | 36.9 |
| HuatuoGPT-Vision (Chen et al., 2024a)-8B | 61.0 | 58.8 | 50.0 | 44.7 | 38.7 | 49.1 |
| *Finetuned VLM* | | | | | | |
| Qwen2.5-VL (Bai et al., 2025)-7B (SFT) | 46.4 | 46.7 | 40.0 | 55.6 | 50.0 | 51.7 |
| Qwen2.5-VL (Bai et al., 2025)-7B (GRPO) | 57.1 | 66.7 | 30.0 | 70.4 | 63.3 | 57.2 |
| MedCCO-7B | 53.6 | 63.3 | 40.0 | 74.1 | 66.7 | **59.3** |

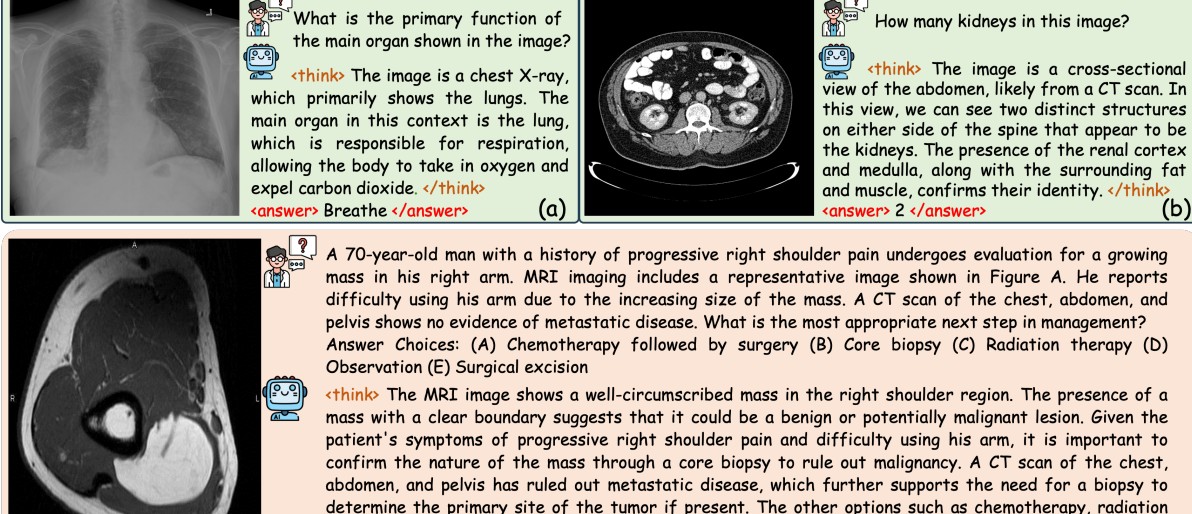

Figure 4: Qualitative results of open-ended and close-ended VQA reasoning. (a) and (b) show open-ended VQA without an option list, while (c) illustrates a close-ended VQA with an option list.

**Training Dynamics.** As shown in Fig. 5, at Stage 1, the model exhibits weak instruction following. During the early stages of training, it fails to adhere to the required output format specified in Table 1. The reasoning is shallow and inconsistent, often missing essential structural elements of the prompt. At Stage 2, the model begins to follow the prescribed format more closely, but it produces overly verbose and redundant responses. While the structure improves, the reasoning remains unfocused and inefficient. At Stage 3, the model enters the reasoning shrinkage phase. As training progresses, it starts to reduce unnecessary content and, for relatively simple close-ended questions, provides direct answers with minimal reasoning, indicating improved confidence and calibration. Finally, at Stage 4, with continued reinforcement training, the model reaches the reasoning incentivization phase. It gradually increases its use of step-by-step reasoning before producing the final answer. Response length increases to a peak and then fluctuates as the model stabilizes. This stage reflects that the model's reasoning capability is fully activated and consistently utilized.

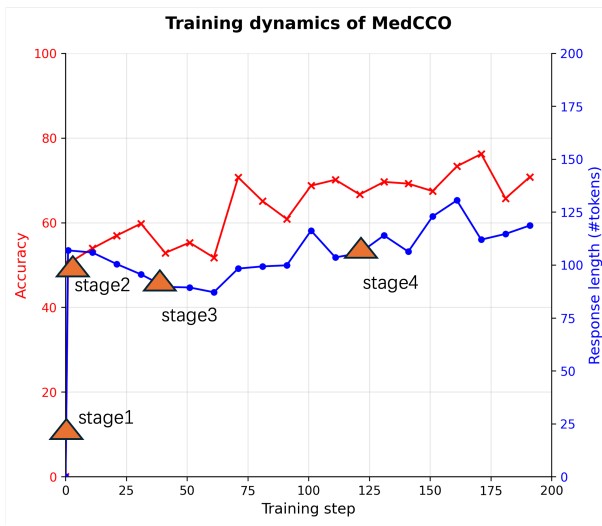

Figure 5: Training dynamics od MedCCO.

## 4.2 ABLATION STUDY

To comprehensively analyze MedCCO's important components, we perform an ablation study on model size, VQA refinement, and fine-tuning type. As for a short demonstration, we report the average performance of OmniMedVQA (Hu et al., 2024), MedXpertQA (Zuo et al., 2025), and MMMU (Yue et al., 2024) Health& Medicine. **Ablations of model architecture/backbone, $\lambda$ and comprehensive comparison of OmniMedVQA (Hu et al., 2024) can be found in Appendix A.4.**

**Model size.** To investigate the impact of model scale on MedCCO, we compared two parameterizations (3B and 7B). As shown in Table 4, the 7B model consistently outperforms its 3B counterpart under identical settings, demonstrating superior capacity. Moreover, Curriculum GRPO yields uniform gains on the in-domain test sets (3.9% for 3B, 2.0% for 7B) and out-of-domain test sets (2.9% for 3B, 5.2% for 7B), underscoring its robustness.

**VQA refinement.** To assess the impact of VQA consistency refinement for open-ended VQA, we conduct joint GRPO experiments using both the close-ended and w/wo refined open-ended data. Figure 3 and Table 4 show that the 3B model improves by 2.5% on in-domain and 3.1% on out-of-domain test sets, while the 7B model achieves gains of 3.9% and 1.2%, respectively. These results confirm the effectiveness of our VQA-consistency optimization.

**Fine-tuning Type.** As shown in Table 4, GRPO-based fine-tuning consistently outperforms SFT across out-of-domain test sets, demonstrating superior generalization capability. While SFT performs slightly better on in-domain evaluations, we attribute this to its strength in learning input-output mappings within the training distribution. In contrast, GRPO encourages reasoning and pathway exploration over direct mapping, making it more robust for unfamiliar or out-of-distribution scenarios.

**Curriculum generalizes better than joint way.** Table 4 shows consistent performance gains on both in-domain and out-of-domain test sets across 3B and 7B models. We attribute this improvement to the mitigation of reward conflict between close-ended and open-ended tasks. Specifically, close-ended tasks adopt discrete rewards, where small differences in output can lead to large variations in reward signals, resulting in sharper gradient updates. In contrast, open-ended tasks use continuous rewards, which produce smoother and less sensitive gradients. When trained jointly, this discrepancy causes gradient imbalance that hinders stable learning. Curriculum learning alleviates this issue by first training the model on close-ended tasks to build a robust reasoning foundation, then gradually adapting it to open-ended tasks. This staged approach bypasses the gradient conflict and facilitates stable acquisition of open-ended reasoning skills. Additionally, KL divergence regularization penalizes policy deviation, further promoting steady and consistent policy improvement.

## 5 CONCLUSION

In this paper, we introduce MedCCO, the first multi-modal medical reasoning model capable of jointly addressing close-ended and open-ended VQA tasks within a unified framework. We introduce a novel hybrid reward function

Table 4: Ablation study on model size, VQA refinement, and fine-tuning type. SFT: supervised fine-tuning; J.: joint; Quilt.: Quilt-VQA; PMC.: PMC-VQA; MedXpert.: MedXpertQA; OmniMed.: OmniMedVQA; H&M: Health and Medical track of MMMU.

| Model Size | VQA refine | FT type | In-domain test | | | | Out-of-domain test | | | | | |
|---|---|---|---|---|---|---|---|---|---|---|---|---|
| | | | VQA-RAD | SLAKE | PathVQA | Avg | Quilt. | PMC. | MedXpert. | OmniMed. | H&M | Avg |
| 3B | - | - | 62.2 | 68.3 | 62.4 | 64.3 | 53.4 | 48.4 | 20.3 | 61.6 | 53.8 | 47.5 |
| | ✓ | SFT | 71.7 | 81.7 | 87.5 | 80.3 | 51.6 | 49.9 | 21.6 | 61.7 | 51.7 | 47.3 |
| | ✗ | J.GRPO | 64.5 | 72.4 | 83.4 | 73.4 | 62.7 | 49.1 | 20.6 | 59.3 | 49.7 | 48.3 |
| | ✓ | GRPO | 66.5 | 73.6 | 81.1 | 73.7 | 63.5 | 47.0 | 22.0 | 61.7 | 51.0 | 49.0 |
| | ✓ | J.GRPO | 66.5 | 79.3 | 81.9 | 75.9 | 65.9 | 53.0 | 20.8 | 63.0 | 54.5 | 51.4 |
| | ✓ | MedCCO | 69.7 | 80.5 | 82.5 | 77.6 | 62.9 | 53.1 | 22.7 | 64.6 | 56.6 | 51.9 |
| 7B | - | - | 67.3 | 71.6 | 65.5 | 68.1 | 54.8 | 50.4 | 21.9 | 63.5 | 51.7 | 48.5 |
| | ✓ | SFT | 71.3 | 78.6 | 87.8 | 79.2 | 60.9 | 49.2 | 20.3 | 55.7 | 51.7 | 47.6 |
| | ✗ | J.GRPO | 69.3 | 67.8 | 83.7 | 73.6 | 63.1 | 52.2 | 22.2 | 63.5 | 57.9 | 51.8 |
| | ✓ | GRPO | 70.5 | 79.3 | 82.8 | 77.5 | 50.2 | 51.2 | 21.4 | 65.1 | 57.2 | 49.0 |
| | ✓ | J.GRPO | 72.1 | 75.7 | 84.7 | 77.5 | 67.2 | 53.6 | 22.0 | 63.6 | 58.6 | 53.0 |
| | ✓ | MedCCO | 76.3 | 79.4 | 82.8 | 79.5 | 69.4 | 53.2 | 23.4 | 65.8 | 59.3 | 54.2 |

that integrates lexical and semantic metrics to jointly optimize textual fluency and semantic coherence, which proves effective for open-ended RFT. By adopting a curriculum training strategy that transitions from structured to open-ended supervision, MedCCO enhances reasoning performance while maintaining training stability, achieving state-of-the-art results across diverse medical benchmarks. More broadly, this work sets a foundation for developing clinically adaptable AI systems that support flexible, context-aware reasoning, moving beyond rigid answer formats toward more nuanced and human-aligned medical understanding.

This study represents an initial attempt to integrate both close-ended and open-ended medical VQA data within a GRPO training framework to enhance the reasoning capabilities of medical VLMs. Our experiments reveal that reinforcement learning with open-ended data presents greater challenges compared to training on close-ended questions with predefined answer options. In particular, automatically evaluating the quality of open-ended answers is particularly challenging in the medical domain during RFT. While employing powerful LLMs as judge is a promising alternative, it introduces concerns regarding cost, latency, and potential evaluation bias. As a practical compromise, our work utilizes NLP metrics and BERTScore to balance textual fluency and semantic coherence, though this approach may not fully capture fine-grained clinical correctness. Future work should explore better trade-offs between evaluation accuracy and efficiency, potentially through more cost-effective reward models.

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

## A APPENDIX

### A.1 ETHICAL CONSIDERATIONS

The present research did not utilize any private or sensitive data, nor did it involve human participants, animal subjects, or personally identifiable information. All experimental data were sourced from publicly available datasets. The methodologies, analyses, and outcomes described herein do not present any foreseeable ethical risks or potential for harm. This work has been conducted in strict adherence to the ICLR Code of Ethics and follows all applicable ethical guidelines for responsible research.

### A.2 REPRODUCIBILITY

We have taken comprehensive measures to support the reproducibility of our experiments. Detailed descriptions of the model architectures, hyperparameter settings, and training protocols are included in Section 4. To enable replication and verification of our results, we provide the full source code, along with configuration files and usage instructions, in an anonymized repository available at: https://anonymous.4open.science/r/MedCCO/.

### A.3 THE USE OF LARGE LANGUAGE MODELS

Large language models (LLMs) were only used to polish the language of this manuscript. They assisted in improving clarity, style, and grammar, without contributing to the research design, analysis, or conclusions.

### A.4 SUPPLEMENTAL EXPERIMENTAL RESULTS

Table 5: Ablation results on multi-medical VQA benchmarks using Huatuo and Qwen backbones. Highest value per column is marked in red, and second-highest in blue.

| Model | VQA-RAD | SLAKE | Path-VQA | Avg. | Quilt. | PMC. | MedXpert. | OmniMed. | H&M | Avg. |
|---|---|---|---|---|---|---|---|---|---|---|
| Qwen2.5VL-7B-Instruct | 67.3 | 71.6 | 65.5 | 68.1 | 54.8 | 50.4 | 21.9 | 63.5 | 51.7 | 48.5 |
| HuatuoGPT-Vision-8B | 63.8 | 74.5 | 79.9 | 72.7 | 63.9 | 52.7 | 21.7 | 75.1 | 49.1 | 52.5 |
| Qwen2.5VL-7B (SFT) | 71.3 | 78.6 | 87.8 | 79.2 | 60.9 | 49.2 | 20.3 | 55.7 | 51.7 | 47.6 |
| HuatuoGPT-Vision-8B (SFT) | 73.2 | 80.2 | 88.6 | 80.7 | 62.4 | 50.7 | 21.9 | 79.3 | 55.6 | 54.0 |
| MedCOO-Qwen2.5VL-7B | 76.3 | 79.4 | 82.8 | 79.5 | 69.4 | 53.2 | 23.4 | 65.8 | 59.3 | 54.2 |
| MedCOO-HuatuoGPT-Vision-8B | 79.2 | 79.0 | 81.5 | 79.9 | 71.2 | 56.1 | 23.5 | 80.7 | 61.4 | **58.6** |

**RFT with Different Architecture/Backbone.** For a comprehensive assessment of how different visual language model (VLM) architectures adapt to the medical domain, we compared the general zero-shot model Qwen2.5-VL-7B-Instruct and the medically-specialized model HuatuoGPT-Vision-8B after Supervised Fine-Tuning (SFT) and Reinforcement Learning (RL) training. The results shown in Fig. 5 confirm that medical VLMs (with HuatuoGPT-Vision as the backbone) generally outperform general zero-shot VLMs across most benchmarks, and both SFT and RL training lead to substantial further performance gains. Specifically, in their base versions, HuatuoGPT-Vision-8B achieved a higher average score (72.7%) on the VQA-RAD, SLAKE, and Path-VQA datasets compared to Qwen2.5-VL-7B (68.1%), indicating the advantage of its inherent medical prior knowledge. After SFT, both models showed improved performance, with HuatuoGPT-Vision-8B (SFT) reaching an average score of 80.7% on medical VQA tasks, surpassing Qwen2.5-VL-7B (SFT)'s 79.2%. Further RL with our MedCCO yielded additional improvements. The MedCOO-tuned HuatuoGPT-Vision-8B model demonstrated more robust performance on comprehensive medical benchmarks (e.g., Quilt, PMC, OmniMed), achieving an average score of 58.6%, which is significantly higher than the 54.2% achieved by MedCOO-tuned Qwen2.5-VL-7B. These results validate that a medically-specialized architecture as the backbone, when combined with SFT and RL optimization, more effectively captures medical semantics and enhances the model's robustness and accuracy in complex healthcare scenarios. Additionally, we have incorporated the latest Qwen3-VL-8B as our backbone model. The results demonstrate a consistent performance improvement over the GRPO baseline (avg 58.0 → 61.6), highlighting the significant effectiveness of both open-ended RFT and the curriculum learning regime.

**Ablation Study of $\lambda$.** From the ablation study results in Table 7, we can observe two key findings: (1) MedCCO exhibits strong robustness across different $\lambda$ values, with balanced weighting between lexical and semantic metrics (e.g., $\lambda = 0.5$ or 0.7) achieving optimal performance; (2) Exclusive reliance on either metric type degrades results, though semantic metrics demonstrate greater importance (as evidenced by the superior performance of $\lambda = 0.0$ compared to $\lambda = 1.0$). These results establish that approximate equal weighting of both metrics serves as an effective practical guideline for $\lambda$ selection.

Table 6: Generalization performance across diverse VLM backbones. Best results are highlighted in bold.

| Model | Quilt. | PMC. | MedXpert. | OmniMed. | H&M | Avg |
|---|---|---|---|---|---|---|
| Qwen2.5-VL-3B | 53.4 | 48.4 | 20.3 | 61.6 | 53.8 | 47.5 |
| Qwen2.5-VL-3B(SFT) | 51.6 | 49.9 | 21.6 | 61.7 | 51.7 | 47.3 |
| Qwen2.5-VL-3B(GRPO) | **63.5** | 47.0 | 22.0 | 61.7 | 51.0 | 49.0 |
| MedCCO-3B | 62.9 | **53.1** | **22.7** | **64.6** | **56.6** | **52.0** |
| Qwen2.5-VL-7B | 53.4 | 48.4 | 20.3 | 61.6 | 53.8 | 47.5 |
| Qwen2.5-VL-7B(SFT) | 60.9 | 49.2 | 20.3 | 55.7 | 51.7 | 47.6 |
| Qwen2.5-VL-7B(GRPO) | 50.2 | 51.2 | 21.4 | **65.1** | 57.2 | 49.0 |
| MedCCO-7B | **69.4** | **53.2** | **23.4** | 65.8 | **59.3** | **54.2** |
| HuatouGPT-Vision-8B | 63.9 | 52.7 | 21.7 | 75.1 | 49.1 | 52.5 |
| HuatouGPT-Vision-8B(SFT) | 62.4 | 50.7 | 21.9 | 79.3 | 55.6 | 54.0 |
| HuatouGPT-Vision-8B(GRPO) | 65.3 | 52.6 | 21.2 | 79.4 | 57.1 | 55.1 |
| MedCCO-HuatouGPT-Vision-8B | **71.2** | **56.1** | **23.5** | **80.7** | **61.4** | **58.6** |
| Qwen3-VL-8B | 53.9 | 54.6 | 24.2 | 75.7 | 57.2 | 53.1 |
| Qwen3-VL-8B(SFT) | 52.4 | 55.6 | 23.5 | 78.9 | 62.5 | 54.6 |
| Qwen3-VL-8B(GRPO) | 59.5 | 61.4 | 23.3 | 83.5 | 61.9 | 58.0 |
| MedCCO-Qwen3-VL-8B | **62.4** | **66.3** | **24.9** | **85.7** | **68.6** | **61.6** |

Table 7: Ablation on $\lambda$ across in-domain and out-of-domain test sets. The highest value per column is highlighted in red, and the second-highest in blue.

| $\lambda$ | In-domain test | | | | Out-of-domain test | | | | | |
|---|---|---|---|---|---|---|---|---|---|---|
| | VQA-RAD | SLAKE | PathVQA | Avg | Quilt. | PMC. | MedXpert. | OmniMed. | H&M | Avg |
| 0.0 | 74.5 | 77.0 | 80.1 | 77.2 | 69.0 | 52.5 | 22.8 | 65.0 | 57.5 | 53.4 |
| 0.1 | 75.2 | 77.8 | 81.0 | 78.0 | 69.2 | 51.9 | 23.3 | 64.7 | 58.3 | 53.5 |
| 0.3 | 75.8 | 78.5 | 81.7 | 78.7 | 68.1 | 52.3 | 22.7 | 65.6 | 59.1 | 53.6 |
| 0.5 | 76.4 | 79.3 | 83.1 | 79.6 | 69.5 | 53.8 | 24.1 | 65.4 | 59.0 | 54.4 |
| 0.7 | 76.3 | 79.4 | 82.8 | 79.5 | 69.4 | 53.2 | 23.4 | 65.8 | 59.3 | 54.2 |
| 0.9 | 75.0 | 78.0 | 81.5 | 78.2 | 68.2 | 52.3 | 22.8 | 64.5 | 58.6 | 53.3 |
| 1.0 | 73.5 | 76.2 | 79.4 | 76.4 | 67.9 | 52.6 | 22.3 | 64.8 | 57.5 | 53.0 |

**Performance Comparison using LLM-as-judge.** While BLEU, ROUGE, and BERTScore remain widely adopted in text generation research, we recognize their limitations in assessing clinical relevance during open-ended medical reasoning tasks. To address this, we augmented our evaluation framework with an LLM-as-judge approach using GPT-4o. The specific prompt used for this evaluation is presented in Table 9. Under this rigorous clinical evaluation, MedCCO-7B achieves state-of-the-art performance across all medical VQA benchmarks (Table 8), obtaining the highest scores on VQA-RAD (35.7%), SLAKE (56.8%), and Quilt-VQA (17.8%), with a leading average score of 32.8%. It significantly outperforms both general zero-shot VLMs (e.g., LLaVA-v1.6-34B, Qwen2.5-VL-7B) and medical zero-shot models (e.g., RadFM-13B, HuatuoGPT-Vision-8B), demonstrating its superior effectiveness and clinical reliability in medical visual question answering.

**Performance Comparison on OmniMedVQA (Hu et al., 2024) Dataset.** We provide additional experiments on OmniMedVQA (Hu et al., 2024) dataset. As illustrated in Table 10, For OmniMedVQA (Hu et al., 2024), our MedCCO attains a competitive second-best result of 65.8% (averaged accuracy), surpassing all other models except HuatuoGPT-Vision (Chen et al., 2024a)-8B(75.1%). The surprisingly high results of HuatuoGPT-Vision (Chen et al., 2024a)-8B were achieved by leveraging over 1 million curated medical images, which comprise almost all the public datasets as described in (Chen et al., 2024a) (largely overlapping with OmniMedVQA (Hu et al., 2024)), while MedCCO is trained with only around 27k medical question-answer pairs (3% of the counterpart and no overlapping with OmniMedVQA (Hu et al., 2024)), highlighting its remarkable data efficiency and consistent performance across diverse modalities. Also, experimental results show that GRPO-based training method offers limited gains for tasks focused purely on visual perception.

Table 8: LLM-as-judge results on medical VQA benchmarks. Best results are highlighted in bold.

| Model | VQA-RAD | SLAKE | Path-VQA | Quilt-VQA | Avg. |
|---|---|---|---|---|---|
| *General Zero-shot VLM* | | | | | |
| Yi-VL-34B | 66.2 | 65.1 | 73.8 | 53.2 | 64.6 |
| LLaVA-v1.6-7B | 65.3 | 68.4 | 78.0 | 55.7 | 66.9 |
| LLaVA-v1.6-13B | 68.4 | 71.9 | 70.5 | 60.6 | 67.9 |
| LLaVA-v1.6-34B | 75.1 | 72.5 | 76.4 | 60.8 | 71.2 |
| Qwen2.5-VL-7B | 76.5 | 70.4 | 78.5 | 62.9 | 72.1 |
| *Medical Zero-shot VLM* | | | | | |
| Med-Flamingo-7B | 74.8 | 63.0 | 83.4 | 60.7 | 70.5 |
| RadFM-13B | 73.3 | 75.0 | 82.5 | 59.2 | 72.5 |
| LLaVA-Med-7B | 75.1 | 69.7 | 85.6 | 65.5 | 74.0 |
| HuatuoGPT-Vision-8B | 83.0 | 83.5 | 86.8 | 66.8 | 80.0 |
| *Finetuned VLM* | | | | | |
| Qwen2.5-VL-7B (SFT) | 76.5 | 79.8 | 80.2 | 64.5 | 75.3 |
| Qwen2.5-VL-7B (GRPO) | 77.8 | 76.8 | 81.4 | 66.6 | 75.7 |
| MedCCO-7B | 84.9 | 85.6 | 88.9 | 70.5 | 82.5 |

Table 9: Prompt used for LLM-as-judge evaluation.

| Prompt |
|---|
| As a medical expert, evaluate the following: Question: {question}; Predicted Answer: {pred_answer}; Correct Answer: {gt_answer}. Evaluate: Is the predicted answer clinically equivalent to the correct answer? Consider: 1. Medical correctness 2. Clinical relevance 3. Equivalent meaning (ignore wording differences). Respond ONLY with "True" or "False". |

## A.5 DETAILS OF DATASETS

For our experiments, we employ eight open-source medical VQA datasets. The details are provided below:

**VQA-RAD** (Lau et al., 2018) is a radiology-focused medical VQA dataset consisting of 3,064 question-answer (QA) pairs for training and 451 for testing. Among the training samples, there are 1,242 open-ended and 1,823 close-ended questions. The test set contains 272 close-ended and 179 open-ended questions.

**SLAKE** (Liu et al., 2021) is a bilingual English-Chinese dataset; we use only the English portion in our experiments. It includes data from three imaging modalities: X-ray, CT, and MRI. After removing Chinese entries, the dataset comprises 4,919 training samples (2,976 open-ended, 1,943 close-ended), 1,053 validation samples (657 open-ended, 422 close-ended), and 1,061 test samples (671 open-ended, 416 close-ended).

**PathVQA** (He et al., 2020) is designed for pathology-related visual question answering, including pathological images, cellular microscopy, and some natural disease-related images. The dataset contains 19,654 training samples (9,553 open-ended, 10,621 close-ended), 6,259 validation samples (2,967 open-ended, 3,435 close-ended), and 6,719 test samples (3,201 open-ended, 3,659 close-ended).

**Quilt-VQA** (Seyfioglu et al., 2024) is a benchmark for pathology visual question answering (VQA), specifically focusing on pathology images. It comprises 344 close-ended and 957 open-ended VQA pairs for evaluation.

**PMC-VQA** (Zhang et al., 2023) contains approximately 227K VQA pairs and 149K medical images, covering a broad spectrum of diseases and radiological modalities. For evaluation, we adopt a subset consisting of 2,000 test examples selected from the validation split of HuatuoGPT-Vision (Chen et al., 2024a).

**MedXpertQA** (Zuo et al., 2025) is a recently introduced benchmark designed to assess the complex medical reasoning capabilities of large language models (LLMs) and vision-language models (VLMs) using expert-level questions. It comprises two subsets: a pure-text QA set and a multi-modal (MM) VQA set. In our experiments, we use only the MM subset, which includes 2,000 multiple-choice questions.

**MMMU Health & Medicine Track** (Yue et al., 2024) is part of the broader MMMU benchmark and is aimed at evaluating the performance of multi-modal models across a wide range of medical tasks. It covers multiple subfields, including Basic Medical Science, Clinical Medicine, Diagnostics and Laboratory Medicine, Pharmacy, and Public Health. The test set includes 150 multiple-choice questions, as used in (Chen et al., 2024a).

**OmniMedVQA** (Hu et al., 2024) is constructed from 41 classical medical imaging tasks, reformulated for multi-modal evaluation. It is designed to comprehensively assess the visual understanding capabilities of multi-modal models in medical imaging. We use the test split adopted in (Chen et al., 2024a), which contains 11,124 examples.

### A.6 VQA CONSISTENCY REFINEMENT

**Prompt.** Before training on the open-ended data, we employ Qwen2.5-VL (Bai et al., 2025)-72B as an auditor to refine the consistency of our VQA data. The objective of this refinement step is to guide the model in understanding what to answer, rather than merely learning question-answer mappings during supervised fine-tuning (SFT). We apply this consistency refinement process to the VQA-RAD (Lau et al., 2018), SLAKE (Liu et al., 2021) (English portion), and PathVQA (He et al., 2020) datasets. Figure 6 illustrates the detailed prompt used in this process.

**VQA refinement examples.** Table 11 presents three examples illustrating our VQA consistency refinement process. After refinement, the alignment between the question and the answer is significantly improved, which helps the model focus on understanding what to answer rather than merely learning superficial mappings. For additional examples of VQA refinement, please refer to our code.

### A.7 ADDITIONAL QUALITATIVE RESULTS

**VQA-Consistency Auditor Prompt**

```
ori_q: {Original Question}
ori_a: {Answer}
```
**Role: QA-Consistency Auditor** – an expert data-curator.

Your task is to refine **open-ended visual-question-answering (VQA)** pairs so that the revised question and answer remain logically and granularly consistent. These are **open-end VQA pairs**, not closed-end: do **not** embed answer choices in the question.

**Process:**

1. Read the original question (`ori_q`).

2. Ignore the visual content; focus only on the wording of the question and the expected form of the answer.

3. Internally simulate an expert's likely free-form answer (`Expert_Guess`).

4. Compare `Expert_Guess` to the original answer (`ori_a`) to spot missing components or granularity gaps.

5. Decide on a status:
   - **consistent** – `ori_q` already elicits exactly the information found in `ori_a`.
   - **needs_fix** – `ori_q` is too broad, ambiguous, or does not explicitly request every element found in `ori_a`.
   - **drop** – The pair is unusable (contradictory, nonsensical, etc.).

6. If the status is `needs_fix`, craft `new_q` that:
   - Starts with a precise **action verb** ("Identify", "Describe", "Explain", …).
   - Explicitly requests **every component** required by `ori_a`.
   - Maintains an **open-end** format (no yes/no phrasing, no embedded choices).
   - Provides a **1-to-1 mapping**: each phrase in `ori_a` must correspond to a clearly stated element in `new_q`.
   - Matches the **granularity** of `ori_a` exactly—no more, no less.
   - Ensures `new_a` presents components in the **same order** that `new_q` requests them.

7. Adjust `new_a` only if wording changes are necessary for brevity or clarity; never change the meaning.

**Key Requirements:**
- **Open-ended:** Questions must allow free-form expert responses; never embed answer choices.
- **Multi-component precision:** If the answer contains multiple elements, the question must explicitly ask for each.
- **Action-verb prompts:** Begin revised questions with verbs like "Identify", "Describe", "Explain".
- **Granularity match:** Question scope must match answer specificity exactly.
- **Order consistency:** Arrange components in `new_a` in the same sequence as requested in `new_q`.
- **Answer conciseness:** Keep `new_a` as short as possible while fully capturing the meaning.

**Output format:**

Return **one** JSON object—nothing else—using this template:

```
{
  "status": "consistent | needs_fix | drop",
  "ori_q": "<string>",
  "ori_a": "<string>",
  "new_q": "<string>",
  "new_a": "<string>",
  "notes": "<less than 15 words rationale>"
}
```

Figure 6: Prompt for refining open-ended VQA consistency.

Table 10: Generalization (zero-shot) comparison on the OmniMedVQA (Hu et al., 2024) benchmark across different imaging modalities. Results are reported at the modality level of accuracy. Abbreviations: CT: Computed Tomography; FP: Fundus Photography; MRI: Magnetic Resonance Imaging; OCT: Optical Coherence Tomography; Der: Dermoscopy; Mic: Microscopy; US: Ultrasound.

| Model | CT | FP | MRI | OCT | Der | Mic | X-Ray | US | Avg. |
|---|---|---|---|---|---|---|---|---|---|
| *General VLM* | | | | | | | | | |
| Yi-VL (Young et al., 2024)-34B | 39.8 | 57.2 | 51.4 | 70.5 | 54.5 | 61.4 | 64.2 | 40.5 | 54.9 |
| LLaVA-v1.6 (Liu et al., 2024)-7B | 40.1 | 39.5 | 54.8 | 58.4 | 54.0 | 48.8 | 53.3 | 47.9 | 49.6 |
| LLaVA-v1.6 (Liu et al., 2024)-13B | 40.0 | 43.6 | 47.4 | 63.2 | 58.0 | 59.6 | 42.6 | 50.6 | |
| LLaVA-v1.6 (Liu et al., 2024)-34B | 50.6 | 63.4 | 60.9 | 68.4 | 65.7 | 62.8 | 74.7 | 44.5 | 61.4 |
| LLaVA-v1.5-LLaMA3-8B | 33.0 | 49.7 | 53.8 | 76.0 | 63.1 | 48.4 | 56.6 | 31.2 | 48.8 |
| Qwen2.5-VL (Bai et al., 2025)-7B | 68.9 | 78.6 | 56.3 | 64.4 | 66.5 | 68.8 | 75.6 | 29.1 | 63.5 |
| *Medical VLM* | | | | | | | | | |
| Med-Flamingo (Moor et al., 2023) | 34.6 | 33.3 | 27.5 | 26.0 | 28.3 | 28.1 | 30.1 | 33.2 | 30.2 |
| RadFM (Wu et al., 2023) | 33.3 | 35.0 | 22.0 | 31.3 | 36.3 | 28.0 | 31.5 | 26.1 | 30.5 |
| LLaVA-Med (Li et al., 2023b)-7B | 25.3 | 48.4 | 35.9 | 42.1 | 45.2 | 44.0 | 31.7 | 83.7 | 44.5 |
| HuatuoGPT-Vision (Chen et al., 2024a)-8B | 61.6 | 80.2 | 65.1 | 86.3 | 71.6 | 67.4 | 81.4 | 87.4 | 75.1 |
| *Finetuned VLM* | | | | | | | | | |
| Qwen2.5-VL (Bai et al., 2025)-7B (SFT) | 48.5 | 57.8 | 56.0 | 67.7 | 60.4 | 52.1 | 66.9 | 36.2 | 55.7 |
| Qwen2.5-VL (Bai et al., 2025)-7B (GRPO) | 69.9 | 79.4 | 58.2 | 70.0 | 70.1 | 67.7 | 75.9 | 29.7 | 65.1 |
| MedCCO-7B | 71.4 | 73.5 | 62.5 | 63.2 | 69.5 | 66.6 | 74.2 | 37.3 | 65.8 |

Table 11: Examples of Refined VQA Questions and Answers

| Field | Content |
|---|---|
| **Example 1** | |
| Original Question | What is the main organ in the image? |
| Revised Question | Identify the main organs visible in the image. |
| Answer | Liver, Heart, Spleen, Lung |
| Notes | Clarifies the need to identify multiple organs. |
| **Example 2** | |
| Original Question | How was this image taken? |
| Revised Question | Identify the imaging modality used to capture this image. |
| Answer | X-Ray |
| Notes | Clarifies the specific imaging technique. |
| **Example 3** | |
| Original Question | What type of imaging is this? |
| Revised Question | Identify the imaging modality and sequence type shown in the image. |
| Answer | MRI, Diffusion Weighted |
| Notes | Clarifies both modality and sequence type. |

Table 12: Open-ended VQA: Example 1.

**Open-ended VQA: Example 1**

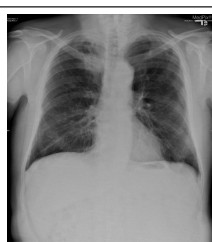

| | |
|---|---|
| **Question** | Identify the lobe where an abnormal mass is visible in the provided image. |
| **Answer** | `<think>` The image is a chest X-ray, which typically shows the lungs, heart, and other structures of the thoracic cavity. In this X-ray, there appears to be a shadow or opacity on the right side of the image, which could indicate an abnormality. The location of this shadow suggests it is in the right lung. The left lung does not appear to have a similar shadow. Therefore, the abnormal mass is likely in the right lower lobe. `</think>` `<answer>` Right upper lobe `</answer>` |

Table 13: Open-ended VQA: Example 2.

**Open-ended VQA: Example 2**

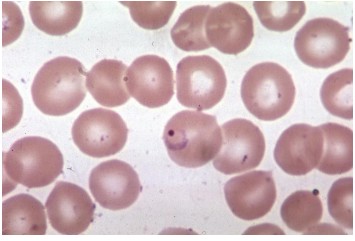

| | |
|---|---|
| **Question** | Identify the specific type of malaria parasite present in the image. |
| **Answer** | `<think>` The image shows red blood cells with a single parasite inside one of them, which is characteristic of Plasmodium vivax malaria. The presence of a single parasite within a red blood cell is indicative of the trophozoite stage of the parasite's life cycle. The shape and color of the parasite are consistent with the appearance of Plasmodium vivax. `</think>` `<answer>` Plasmodium vivax `</answer>` |

Table 14: Close-ended VQA: Example 1.

**Close-ended VQA: Example 1.**

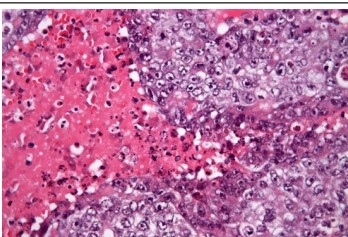

**Question**    A 30-year-old male presents to the primary care clinic for evaluation of a painful right-sided scrotal mass. The patient otherwise has no significant past medical history and does not take any daily medications. He denies direct or indirect trauma to the scrotum. He denies tobacco, alcohol, and illicit drug use. Temperature is 98.6 °F. Blood pressure is 120/80 mmHg, pulse is 80/min, and respiratory rate is 12/min. Palpation of the scrotum demonstrates a unilateral tender mass. Laboratory testing is significant for an elevation in human chorionic gonadotropin and normal alpha-fetoprotein levels. Histopathologic evaluation of the mass is shown in the image below*.* Which of the following is the most accurate regarding the testicular tumor pathology in this patient? Answer Choices: (A) SALL4 marker negativity (B) Orchiectomy should be avoided (C) Worse prognosis than seminoma (D) Rarely appears as a mixed tumor (E) Analogous to dysgerminoma

**Answer**    `<think>` The histopathology shows a pattern consistent with a seminoma, which is a type of testicular cancer. Seminomas are typically characterized by large, round cells with a clear cytoplasm and a prominent nucleolus. The presence of a single type of cell without significant pleomorphism or necrosis is characteristic of seminoma. The elevation in human chorionic gonadotropin (hCG) supports the diagnosis of a germ cell tumor, which includes seminoma. The other options provided do not accurately describe the testicular tumor pathology in this patient based on the given information and the image. `</think>` `<answer>` C `</answer>`

Table 15: Close-ended VQA: Example 2.

**Close-ended VQA: Example 2**

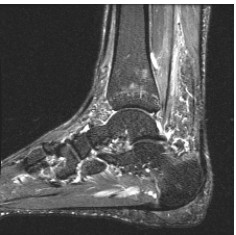

**Question**    What can be observed in this image? A. Shin splints B. Tennis elbow C. Plantar fascia pathology D. Meniscal tear

**Answer**    `<think>` The image appears to be an MRI scan of the foot, which is commonly used to diagnose soft tissue injuries. The presence of fluid or abnormal signal intensity around the plantar fascia suggests inflammation or pathology in that area. This is characteristic of plantar fasciitis, a condition often referred to as p̈lantar fascia pathology.T̈here is no indication of shin splints, tennis elbow, or meniscal tear in this image. `</think>` `<answer>` C `</answer>`

