# OpenReview forum: "MedCCO: Unleashing Open-Ended Reasoning in Medical Multi-modal Language Models via Curriculum Reinforcement Learning"
_ICLR.cc/2026/Conference — Submitted to ICLR 2026_

### Official Review · Reviewer_7Zru · 2025-10-14

**Soundness:** 3
**Presentation:** 3
**Contribution:** 2
**Rating:** 4
**Confidence:** 4

**Summary:**

This paper presents MedCCO, a curriculum reinforcement learning framework for medical visual question answering to handle both closed-ended and open-ended questions. The model is first trained on the closed-ended tasks and then progressively adapts to more complex, open-ended tasks with a hybrid reward function. Experimental results demonstrate improved performance across eight medical VQA benchmarks in both in-domain and out-of-domain tasks.

**Strengths:**

1. This paper proposes a curriculum learning strategy to improve the reasoning ability of medical VLMs on medical VQA. By staging the training, the model avoids the gradient imbalance that would otherwise hinder optimization.
2. The proposed method is evaluated on eight benchmarks, including five out-of-domain datasets. The results demonstrate the effectiveness of the method and its generalization to new domains.
3. The paper is well presented and easy to understand.

**Weaknesses:**

1. The paper claims to address open-ended reasoning (e.g., in the title), but the tasks are still limited in generating very short words or phrases of VQA tasks, which structurally resemble fill-in-the-blank answers rather than true free-text generation. It means the method isn't validated on tasks requiring more complex, narrative reasoning, such as generating diagnostic summaries or radiology report sections.
2. The ablation study on the open-ended reward function feels incomplete. It does not include a direct comparison with a simpler exact-match reward baseline (or is the GRPO row in Table 2 referring to this? but not sure if it's done in a similar curriculum manner). Considering that the ground-truth answers are quite short, an exact-match metric could actually serve as a strong baseline. Without this comparison, it’s hard to tell how much the proposed hybrid reward (based on BLEU, ROUGE, and BERTScore) truly contributes beyond what a straightforward approach might already achieve.
3. I think it would be helpful to compare with (1) state-of-the-art medical models such as MedGemma [1], and (2) medical reasoning models, e.g. [2][3].

[1] MedGemma Technical Report https://arxiv.org/abs/2507.05201 \
[2] MedVLM-R1: Incentivizing Medical Reasoning Capability of Vision-Language Models (VLMs) via Reinforcement Learning \
[3] MEDVLTHINKER: Simple Baselines for Multimodal Medical Reasoning

**Questions:**

1. Since chain-of-thought SFT before RL has been shown to strengthen reasoning performance, it would be interesting to see how the model behaves with and without this initial “cold-start” stage.
2. The authors mentioned combining multiple rewards to prevent reward hacking. But BLEU and ROUGE are still vulnerable to reward hacking because of the unigram matching. Also I'm not sure if BertScore is accurate enough to capture to distinguish between semantically similar yet meaningfully different phrases. Have the authors observed any related phenomena during training?

---

> ### Author Response · Authors · 2025-11-21
> **Part1**
>
> Dear reviewer 7Zru, thank you for your review and valuable suggestions regarding our work. Below, please find our responses to your concerns.
>
> - **Weakness 1-  True Free-text Generation**: We greatly appreciate the reviewer’s valuable feedback regarding the evaluation of open-ended reasoning. In response to this, we have expanded our experiments to include the generation of radiology reports. Specifically, we perform an evaluation on the MIMIC-CXR-2.0 dataset, where we randomly sample 5,000 examples for testing. The evaluation metrics include BLEU, BertScore, ChestXpert Similarity (which measures disease prediction accuracy), and LLM-as-judge (which assesses the alignment between the generated report and the ground truth).
>
>     We have also compared our method with strong baselines, including LLaVA-Med and HuatuoGPT-Vision. The results, as shown in the table below, demonstrate that MedCCO achieves state-of-the-art performance.
>
>     | Methods               | BLEU1  | BLEU2  | BLEU3  | BLEU4  | BertScore | Chest_sim | LLM-as-judge |
>     |-----------------------|--------|--------|--------|--------|-----------|-----------|--------------|
>     | LLaVA-Med             | 0.261  | 0.212  | 0.119  | 0.082  | 0.337     | 0.353     | 51.4         |
>     | HuatuoGPT-Vision-8B   | 0.311  | 0.235  | 0.125  | 0.094  | 0.353     | 0.349     | 54.6         |
>     | Qwen2.5VL-Instruct-7B | 0.299  | 0.22   | 0.121  | 0.099  | 0.34      | 0.358     | 53.9         |
>     | Qwen2.5VL-Instruct-7B-GRPO                  | 0.315  | 0.231  | 0.128  | 0.102  | 0.364     | 0.351     | 55.2         |
>     | MedCCO                | **0.353**  | **0.261**  | **0.163**  | **0.112**  | **0.415**     | **0.417**     | **62.5**         |
>
>     Furthermore, our findings indicate that using only close-ended data for RFT in tasks such as report generation, which require long-chain reasoning, provides limited improvement. This reinforces the importance of utilizing open-ended data for RFT, further validating the significance of our approach.

---

> ### Author Response · Authors · 2025-11-21
> **Part2**
>
> - **Weakness 2-  Comparison with Exact-match Baseline**: Thank you for your valuable suggestion. We have now included the comparison results with the exact-match based open-ended reward below. The results show only limited performance improvement when compared to the GRPO baseline (Avg 59.7 → 60.0). In contrast, using open-ended metrics (e.g., our proposed open-ended reward) yields a much more significant performance improvement (59.7 → 63.7). This highlights the importance of both lexical and semantic alignment between the generated answer and the ground truth. Despite the relatively short length of the ground truth answers (average of 5 words), it is challenging for the model to produce an exact match, leading to lower reward scores. To address this, we introduce recall and precision measures for both lexical and semantic evaluations, which relax the strict constraint of exact matching, thus making the training process more efficient.
>
>     | Method                     | VQA-RAD | SLAKE | Path-VQA | Quilt. | PMC.  | MedXpert. | OmniMed. | H&M   | Avg   |
>     |----------------------------|---------|-------|----------|--------|-------|-----------|----------|-------|-------|
>     | GRPO                        | 70.5    | 79.3  | 82.8     | 50.2   | 51.2  | 21.4      | 65.1     | 57.2  | 59.7  |
>     | Joint-GRPO                  | 72.1    | 75.7  | 84.7     | 67.2   | **53.6**  | 22.0      | 63.6     | 58.6  | 62.2  |
>     | medcco(exact match)         | 72.5    | **80.5**  | 81.1     | 50.4   | 51.4  | 22.8      | 62.5     | 58.4  | 60.0  |
>     | medcco(open-ended metrics)  | **76.3**    | 79.4  | **82.8**     | **69.4**   | 53.2  | **23.4**      | **65.8**     | **59.3**  | **63.7**  |
>
>
>
>     Additionally, we present the detailed open-ended results and the LLM-as-judge evaluation for a more comprehensive clinical reasoning test (Quilt-VQA dataset). The results further demonstrate the consistent effectiveness of using open-ended metrics rather than an exact-match based reward.
>
>     | Model                         | bert_score | BLEU1  | ROUGE1 | overall | LLM-as-judge |
>     |-------------------------------|------------|--------|--------|---------|--------------|
>     | Yi-VL-34B                      | 0.1456     | 0.0954 | 0.1578 | 0.1323  | 53.2          |
>     | LLaVA-v1.6-7B                  | 0.0967     | 0.0614 | 0.1054 | 0.0874  | 55.7          |
>     | LLaVA-v1.6-13B                 | 0.3146     | 0.1823 | 0.2493 | 0.2454  | 60.6         |
>     | LLaVA-v1.6-34B                 | 0.2975     | 0.1806 | 0.2417 | 0.2371  | 60.8         |
>     | Qwen2.5-VL-7B                  | 0.3625     | 0.2285 | 0.2883 | 0.2896  | 62.9         |
>     | Med-Flamingo                   | 0.2589     | 0.1687 | 0.2461 | 0.2229  | 60.7          |
>     | RadFM                          | 0.2874     | 0.1643 | 0.2045 | 0.2153  | 59.2          |
>     | LLaVA-Med-7B                   | 0.3648     | 0.2278 | 0.2951 | 0.2925  | 65.5         |
>     | MedCCO(exact match)            | 0.3643     | 0.2367 | 0.2915 | 0.2942  | 67.2         |
>     | MedCCO(open-ended metrics)     | **0.4505**     | **0.3303** | **0.4062** | **0.3929**  | **70.5**         |
>
> - **Weakness 3-  Comparison with SOTA Medical (reasoning) Models**: We appreciate the reviewer’s suggestion to compare our method with stronger medical reasoning VLMs. In response, we have included the results of MedGemma, Med-VL-Thinker, and MedVLM-R1 in our evaluation. The updated results clearly demonstrate that MedCCO consistently delivers competitive, and often superior, performance, further validating the effectiveness of our approach.
>
>     | Model         | VQA-RAD | SLAKE | Path-VQA | Quilt. | PMC. | MedXpert. | OmniMed. | H&M  | Avg  |
>     |---------------|---------|-------|----------|--------|------|-----------|----------|------|------|
>     | MedGemma      | 72.5    | 74.7  | 63.8     | 63.8   | 46.0 | 23.1      | 69.5     | 44.8 | 57.3 |
>     | Med-VL-Thinker| 64.9    | 64.1  | 66.0     | 53.0   | 51.6 | **23.4**    | 57.8     | 45.5 | 53.3 |
>     | MedVLM-R1     | 70.5    | 79.3  | 82.8     | 50.2   | 51.2 | 21.4    | 65.1     | 57.2 | 59.7 |
>     | MedCCO-7B     | **76.3** | **79.4**| **82.8** | **69.4** | 53.2 | **23.4**    | **65.8**     | **59.3** | **63.7** |

---

> ### Author Response · Authors · 2025-11-21
> **Part 3**
>
> - **Question 1-  Comparison w and w/o "cold-start"**: Thank you for your valuable suggestion. We would like to clarify that curriculum learning is employed to balance reward conflicts when unifying the two types of RFT in medical VQA. We do not use close-ended RL as a cold-start for open-ended RL, but rather as part of a unified training regime. However, we appreciate the reviewer’s suggestion to explore the effect of removing close-ended RL (as shown in the second row of the table below) to assess the individual contributions of close-ended RL and open-ended RL.
>
>     | Method            | close-ended data | open-ended data | VQA-RAD | SLAKE | Path-VQA | Quilt. | PMC.  | MedXpert. | OmniMed. | H&M   | Avg   |
>     |-------------------|------------------|-----------------|---------|-------|----------|--------|-------|-----------|----------|-------|-------|
>     | Qwen2.5-VL-7B     | ×                | ×               | 67.3    | 71.6  | 65.5 	  | 54.8   | 50.4  | 21.9      | 63.5     |	51.7  | 55.8  |
>     | close-ended GRPO  | √                | ×               | 70.5    | 79.3  | 82.8     | 50.2   | 51.2  | 21.4      | 65.1     | 57.2  | 59.7  |
>     | open-ended GRPO   | ×                | √               | 69.8    | 77.4  | 80.5     | 45.2   | 48.9  | 20.7      | 62.6     | 55.1  | 57.5  |
>     | Joint GRPO        | √                | √               | 72.1	   | 75.7  | 84.7	  | 67.2   | 53.6  | 22.0 	   | 63.6 	  | 58.6  | 62.2  |
>     | MedCCO            | √                | √               | 76.3    | 79.4  | 82.8     | 69.4   | 53.2  | 23.4      | 65.8     | 59.3  | 63.7  |
>
>   The results show that both open-ended RL and close-ended RL improve performance compared to the Qwen2.5-VL-7B baseline, demonstrating the potential of both RFT types. However, open-ended RL alone performs worse than close-ended RL, likely due to its higher learning difficulty. When the two RFT types are unified through a curriculum approach, we achieve the best performance, highlighting the effectiveness of combining open-ended and close-ended RL into a unified RFT framework for medical VQA.
>
> - **Question 2-  Reward Hacking**: We greatly appreciate the reviewer’s insightful comments and acknowledge the concerns raised regarding BertScore’s ability to distinguish between semantically similar yet meaningfully different phrases. This is exactly why we chose to design a multi-dimensional open-ended reward framework. During training, we did observe that BertScore alone does not always capture subtle distinctions between such phrases. As the results of our ablation study show, when the weight of BertScore (denoted as 1-λ) is set to 1.0, the model performs worse, validating the observation that BertScore on its own can have limitations in this regard.
>
>     $
>     R_{\text{open}}(o, g) = \frac{1}{2} \cdot \lambda \cdot \left( \text{BLEU}_1(o, g) + \text{ROUGE}_1(o, g) \right) + (1 - \lambda) \cdot \text{BERTScore}(o, g)
>     $
>
>     To address this issue, we introduced additional lexical reward metrics, specifically BLEU and ROUGE. As shown in the table below, the incorporation of these metrics effectively mitigates the problem of reward hacking and enhances the performance of the model. We found that combining BertScore with BLEU and ROUGE yields the best performance, highlighting the complementarity of these metrics in improving the quality of generated responses.
>
>
>     |  λ    | VQA-RAD | SLAKE | PathVQA | Avg   | Quilt. | PMC. | MedXpert. | OmniMed. | H&M   | Avg   |
>     |-----|---------|-------|---------|-------|--------|------|-----------|----------|-------|-------|
>     | 0.0 | 74.5    | 77.0  | 80.1    | 77.2  | 69.0   | 52.5 | 22.2      | 65.0     | 57.5  | 53.4  |
>     | 0.1 | 75.2    | 77.8  | 81.0    | 78.0  | 69.2   | 51.9 | 23.3      | 64.7     | 58.3  | 53.5  |
>     | 0.3 | 75.8    | 78.5  | 81.7    | 78.7  | 68.1   | 52.3 | 22.7      | 65.6     | 59.1  | 53.6  |
>     | 0.5 | 76.4    | 79.3  | 83.1    | 79.6  | 69.5   | 53.8 | 24.1      | 65.4     | 59.0  | **54.4**  |
>     | 0.7 | 76.3    | 79.4  | 82.8    | 79.5  | 69.4   | 53.2 | 23.4      | 65.8     | 59.3  | 54.2  |
>     | 0.9 | 75.0    | 78.0  | 81.5    | 78.2  | 68.2   | 52.3 | 22.8      | 64.5     | 58.6  | 53.3  |
>     | 1.0 | 73.5    | 76.2  | 79.4    | 76.4  | 67.9   | 52.6 | 22.8      | 64.8     | 57.5  | 53.0  |
>
>     Furthermore, we agree with the reviewer that using only BLEU and ROUGE might indeed lead to reward hacking due to their reliance on unigram matching. This concern is confirmed in our ablation study, where using BLEU and ROUGE alone (λ = 1) resulted in worst performance. This further emphasizes the need for a multi-dimensional reward strategy, where various evaluation metrics are combined to prevent such issues. Our results reinforce the novelty and effectiveness of our open-ended reward design, which accounts for both lexical and semantic aspects to ensure high-quality and diverse outputs.

---

> > ### Comment · Reviewer_7Zru · 2025-11-25
> >
> > Thank you for your additional experiments and clarifications.
> > I have two more questions regarding the new experiments:
> > - For the open-ended report generation, can you give some statistics of the GT reports and the generation, e.g. number of words, standard variations of the generation etc? I'd like to know how do they change after training.
> > - Could you elaborate the specific experiment setting of the exact match baseline, e.g. how do you combine recall and precision, learning rate, etc?

---

> ### Author Response · Authors · 2025-11-26
>
> Thank you very much for your positive response and constructive questions. We sincerely appreciate the time and effort you have devoted to reviewing our work. We hope that the detailed responses below fully address your remaining concerns.
>
> - **1. Statistics of Ground-Truth and Generated Reports**: We added detailed statistics for the ground-truth (GT) reports and the generated reports for all methods. The last column reports the **average report length in words**, with the **standard deviation** shown in parentheses:
>
>     | Methods| BLEU1 | BLEU2 | BLEU3 | BLEU4 | BertScore | Chest_sim | LLM-as-judge | Report length (words) |
>     |-|-|-|-|-|-|-|-|-|
>     | Ground Truth|–|–|–|–|–|–|–|73.84 (std: 34.13)|
>     | LLaVA-Med| 0.261|0.212|0.119|0.082|0.337|0.353|51.4|167.56 (std: 55.38)   |
>     | HuatuoGPT-Vision-8B|0.311|0.235|0.125|0.094 |0.353|0.349|54.6  |125.23 (std: 42.41) |
>     | Qwen2.5VL-Instruct-7B |0.299|0.220|0.121|0.099|0.340|0.358|53.9|132.85 (std: 37.92)|
>     | GRPO (close-ended RFT)|0.315|0.231|0.128|0.102|0.364|0.351|55.2|116.45 (std: 28.75)|
>     | MedCCO (ours)| **0.353** | **0.261** | **0.163** | **0.112** | **0.415** | **0.417** | **62.5** | **62.43 (std: 15.26)** |
>
>     Compared with the close-ended RFT baseline (GRPO), our joint close-ended + open-ended RFT method MedCCO generates reports whose average length is much closer to the ground truth (62.43 vs. 73.84 words), with a substantially smaller standard deviation (15.26 vs. 34.13 for GT and 28.75 for GRPO), while simultaneously improving all evaluation metrics (BLEU-1–4, BertScore, Chest_sim, and LLM-as-judge). These results show that MedCCO not only enhances the semantic quality of the generated reports but also produces outputs that are better calibrated to the length and style of the ground-truth reports, validating both the effectiveness of our open-ended reward design and the overall significance of our method for radiology report generation.
>
> - **2. Experimental Setting of the Exact-Match Baseline**: In the first rebuttal, the exact-match baseline we reported followed the typical setting used in mathematical reasoning, i.e., a hard-match reward (gt == pred?), so there was no notion of recall or precision. We then realized that it would be helpful to additionally compare rewards that explicitly incorporate recall and precision, and thus in this rebuttal we include experiments based on a reward that combines the two. Specifically, **we use BLEU-1 as precision and ROUGE-1 as recall, with equal weights**. The summarized results are as follows:
>
>     | Method                         | Open-ended reward            | VQA-RAD | SLAKE | Path-VQA | Quilt. | PMC  | MedXpert | OmniMed | H&M  | Avg  |
>     |-|-|-|-|-|-|-|-|-|-|-|
>     | GRPO| – |70.5|79.3|82.8|50.2|51.2|21.4|65.1|57.2|59.7|
>     | Joint-GRPO| BLEU + ROUGE + BERTScore|72.1|75.7|84.7|67.2|53.6|22.0|63.6|58.6| 62.2 |
>     | exact match [hard match]       | pred == gt ?|   72.5  | 80.5 |    81.1  |  50.4  | 51.4 |   22.8 |   62.5 | 58.4 | 60.0 |
>     | exact match [precision+recall] | BLEU + ROUGE| 73.5  | 76.2 |    79.4  |  67.9  | 52.6 |   22.3 |   64.8 | 57.5 | 61.8 |
>     | MedCCO (open-ended metrics) | BLEU + ROUGE + BERTScore| 76.3|79.4| 82.8|69.4| 53.2| 23.4| 65.8|59.3|**63.7**|
>
>     From the results, we can observe that neither the hard-match reward nor the hard matching based on the combination of recall and precision yields performance gains as large as those obtained with our open-ended reward that combines BLEU, ROUGE, and BERTScore. **Although the training answers are relatively short, exact-match-based rewards cannot properly handle cases involving negation**. [Negative error matching in Medical VQA: **In Medical VQA, many correct answers contain negations (e.g., “no evidence of XX”). Simple keyword matching can wrongly penalize such valid responses**, leading to erroneous reward signals (the negation reward problem).] In contrast, **our proposed NLP and semantic based matching metrics provide a more generalizable solution by capturing diverse semantic alignments between generated and reference answers.** Moreover, compared with large reward models, our method achieves faster computation while maintaining competitive effectiveness, as demonstrated in our experimental results.
>
> - **Other Experimental Settings (Hyperparameters)**
>
>     Unless otherwise noted, the training hyperparameters for the exact-match baselines are **identical** to those used for MedCCO. The main settings are:
>
>     | Learning rate | Steps | Global batch size | ppo_mini_batch_size | ppo_micro_batch_size_per_gpu | Max response length | kl_loss_coef | rollout.n |
>     |-|-------|-|-|-|-|-|-|
>     | 1e-6          | 400   | 64 × 10           | 16                  | 10                           | 4096                | 0.01        | 10       |
>
> Once again, we sincerely appreciate the reviewers’ time, effort, and insightful comments, which have greatly helped us improve this work.

---

> > ### Comment · Reviewer_7Zru · 2025-11-26
> >
> > Thank you for the clarification regarding the exact-match baseline.
> > The additional statistics are helpful, so here is the concern: MedCCO’s reports are notably shorter and have a smaller standard deviation. This could imply that (1) shorter reports reduce the error surface, which can inflate BLEU and related metrics; (2) closer alignment with the ground-truth length distribution may mechanically improve lexical or semantic overlap, including LLM-as-judge scores; and (3) the low variance suggests that reports across different images (and potentially different difficulty levels) tend to have similar lengths. In my view, this pattern is not necessarily a negative signal, but it also means these metrics may not be able to demonstrate the model’s effectiveness. This also ties back to my original concern (Weakness 1): generating short words or phrases in a VQA-style setting is a limited task and cannot be taken as evidence of real open-ended reasoning or true report-generation capability.

---

> > > ### Author Response · Authors · 2025-11-27
> > >
> > > Thank you for your insightful questions and thoughtful feedback. We sincerely appreciate the time and effort you have dedicated to reviewing our work. Below, we address your concerns by elaborating on two key aspects:
> > >
> > > ### 1. On the Impact of Report Length
> > > We would like to clarify that the alignment with the ground truth in terms of semantics **is already ensured by BLEU and other metrics**. On top of that, having the report length closer to that of the ground truth is an additional benefit. **In essence, the closer alignment in length is a natural outcome when the semantic content is already aligned. The shorter the response while retaining full information, the better, as it leads to more efficient communication**.
> > >
> > >
> > > Besides, **report length is not a primary evaluation criterion** in radiology report generation. The main objective is to generate **semantically accurate and informative reports**, regardless of their length. Metrics such as BLEU and BERTScore are designed to assess **semantic alignment** and **content coverage**, not merely word count.
> > >
> > > While you raised a valid concern that shorter reports might artificially inflate certain metrics, we emphasize that **modern evaluation metrics inherently account for length effects**. For example:
> > >
> > > -   **BLEU Score**: The BLEU score is specifically designed to handle length discrepancies. It combines n-gram precision with a **Brevity Penalty (BP)**. The n-gram precision $P_{n}$ is computed as the ratio of the total clipped n-gram counts in all candidate translations to the total n-gram counts in the candidates. The Brevity Penalty prevents short candidates from receiving unfairly high scores:
> > >     $$
> > >     \text{BP} =
> > >     \begin{cases}
> > >     1 & \text{if } c > r \\
> > >     \exp(1 - r/c) & \text{if } c \le r
> > >     \end{cases}
> > >     $$
> > >
> > >     where $c$ is the total length of the candidate texts, and $ r $ is the effective reference length. The final BLEU score is the geometric mean of the n-gram precisions, multiplied by the BP:
> > >     $$
> > >     \text{BLEU} = \text{BP} \cdot \exp\left( \sum_{n=1}^N w_n \log P_n \right)
> > >     $$
> > >     **This design ensures that shorter outputs do not unfairly benefit from high n-gram precision, as they are penalized by the BP.**
> > >
> > > Additionally, the **BERTScore** metric evaluates the semantic alignment based on word embeddings rather than the number of words. **BERTScore assesses precision and recall at the semantic level, making it independent of the report's length**. The same principle applies to the **Chest_sim metric**, which **uses a 14-dimensional vector representing common symptoms and negations, and calculates cosine similarity**. Since it is based on symptom presence rather than word count, the report length does not influence this similarity score. For **LLM-as-Judge**, we use GPT-4 to evaluate the semantic alignment between the generated report and the ground truth. **This metric implicitly accounts for both precision and recall** and will give a higher score when the output is semantically consistent with the ground truth, regardless of length.
> > >
> > > In summary, while MedCCO generates reports with lengths closer to the ground truth, **this is not the sole reason for the improvements in BLEU and other metrics**. MedCCO's superior performance **is driven by improved semantic alignment and content coverage, not merely shorter report length**. Despite generating shorter reports, MedCCO maintains high semantic quality and accuracy, as evidenced by metrics like BERTScore, Chest_sim, and LLM-as-Judge. It is important to note that **the task of report generation is not about producing lengthy reports but about providing concise, accurate, and informative outputs**. MedCCO excels in generating reports that are semantically dense and focused, ensuring that critical information is effectively conveyed in a compact form. **From a practical perspective, generating shorter reports that are semantically aligned with the ground truth is indeed desirable**.

---

> > > > ### Author Response · Authors · 2025-11-27
> > > > **Part2**
> > > >
> > > > ### 2. On the Low Variance in Report Length
> > > >
> > > > Regarding the low variance in the report lengths, we argue that this is not a limitation but rather a **strength** of MedCCO. **The low variance indicates that MedCCO generates consistent reports across a range of images or tasks, which is crucial for real-world clinical applications**. In clinical practice, doctors are often required to work with various cases, and they need reliable, consistent reports to make informed decisions quickly.
> > > >
> > > > The consistency in generated report lengths suggests that MedCCO can handle diverse scenarios while maintaining high quality and precision. **This ability to produce consistent reports across different images or tasks is highly desirable in a clinical setting**, where time and accuracy are critical.
> > > >
> > > > Additionally, **the standard deviation (std) in report lengths is not inherently tied to report quality**. If shorter reports dominate the dataset (for example, in cases where short answers account for 90% of the data), the standard deviation will naturally be smaller. This is because **std is a weighted measure of the range and quantity of data points**, and does not directly reflect the semantic quality or effectiveness of the reports. Therefore, the observed low variance does not indicate a flaw in MedCCO’s performance, but rather reflects the **model's ability to generate concise and consistent reports across diverse data**.
> > > >
> > > > In conclusion, the goal of report generation is not to create long, verbose reports, but to deliver accurate and concise diagnostic information. **MedCCO’s ability to generate short, yet information-rich reports aligns with real-world clinical needs, where brevity and accuracy are equally important**.
> > > >
> > > > We hope these clarifications address your concerns. We greatly appreciate your time and constructive feedback.

---

### Official Review · Reviewer_c1rp · 2025-10-28

**Soundness:** 2
**Presentation:** 4
**Contribution:** 2
**Rating:** 2
**Confidence:** 4

**Summary:**

This paper proposes MedCCO, a medical vision language model that can do reasoning for both close- and open-ended visual questions. It employs a curriculum learning paradigm that trains the model first on close-ended questions and then on open-ended questions.  During the open-ended question answering training, the authors propose a hybrid reward function that incorporates both the BLEU/ROUGE and BERTScore to measure the similarity between the hypotheses and the references. The model achieves good performance on both in-domain and out-of-domain benchmarks.

**Strengths:**

1. MedCCO can achieve good performance with reinforcement learning with the hybrid rewarding functions.
2. Some designs on section 3 are interesting. For example, to minimize the affect of different reward scales on close- and open-ended questions, the authors proposed to we-weight the rewards for the GRPO algorithm. They also refine the datasets prior to using them to improve their quality.
3. The paper is well-written with clear descriptions and diverse examples.
4. Thorough ablation studies are conducted in the paper.

**Weaknesses:**

1. The paper lacks novelty on methodology: It is very hard to distinguish this paper from other reasoning papers.  From the viewpoint of methodology, this paper is an adaptation of existing methods to the medical domain.
2. Not truly open-ended: Though the authors claim that MedCCO is curriculum learning based and can answer open-ended questions, its capability is far from real-world open-ended questions.  Datasets like SLAKE have short phrases as the candidate answers, which are not too much different from verifiable answers like multi-choice questions.  They can be easily evaluated with F1 or exact match.
3. Not the first multi-modal medical reasoning model: Despite the ambiguity of the "reasoning," there are some multi-modal medical models, such as MedGemma, which also claimed to have reasoning capability.
4. Quantification of "reasoning" capability: One of the selling point of this paper is "reasoning," while there lack evidence that the model conducts substantial thinking before delivering the final answer except for the case studies. Some quantified metrics, such as the change of response length w.r.t. training steps, can be very helpful.

**Questions:**

See my "Weaknesses" section.

---

> ### Author Response · Authors · 2025-11-21
> **Part1**
>
> Dear Reviewer c1rp, we appreciate your thoughtful review and the constructive suggestions you provided. Below are our responses to the concerns you raised.
>
> - **Weakness 1- Clarify on Our Contribution**: We appreciate the reviewer’s concern and would like to clarify that MedCCO is not a simple adaptation of existing methods. Our framework introduces several methodological elements that have not appeared in prior medical VQA or multimodal RL work. The key contributions are summarized below.
>
>     1. **First Application of Open-Ended Reinforcement Fine-Tuning in Medical VQA**:
>     Previous medical VQA studies focus almost entirely on close-ended formats. We introduce open-ended RFT to this domain for the first time, enabling free-form clinical reasoning rather than classification-style outputs.
>
>     2. **Hybrid Reward Designed for Medical Open-Ended RFT**:
>     We propose a lightweight reward that jointly captures lexical accuracy and clinical semantic correctness. This reward is tailored for medical open-ended RL and also reveals a previously unreported interference between closed-ended and open-ended reward signals.
>
>     3. **Curriculum Learning to Resolve Cross-Task Reward Conflict**:
>     Unlike standard curricula based on data difficulty, our curriculum explicitly addresses the gradient conflict between discrete rewards in close-ended tasks and continuous rewards in open-ended tasks. This conflict arises only when unifying these task types under one RL framework.
>
>     4. **Data Refinement as a Foundational Contribution**:
>     We identify and correct systematic QA granularity issues in public medical VQA datasets. The released refined data strengthens the reliability of training signals and benefits the broader medical VLM community.
>
>
> - **Weakness 2- Not truly open-ended**: We appreciate the reviewer’s feedback and would like to clarify a few points.
>
>     **1. Fact Correction**: The reviewer mentions that the SLAKE dataset contains only short phrases as candidate answers, which resemble multiple-choice questions. However, there is a factual error in this statement. SLAKE actually contains two subsets: one close-ended and one open-ended. The close-ended subset contains candidate answers embedded within the question, whereas the open-ended subset does not. This distinction is crucial as the open-ended subset, which is part of the SLAKE dataset, is designed to evaluate models on more free-form and complex generation tasks, unlike the multiple-choice format.
>
>     **2. Expansion to Real-World Open-Ended Tasks**: To further address the concern regarding the true open-ended capabilities of MedCCO, we have extended our experiments to include the generation of radiology reports, a task that more closely resembles real-world open-ended generation. The average length of each generated report is approximately 175 words, which better aligns with scenarios requiring more complex reasoning. Specifically, we conducted an evaluation on the MIMIC-CXR-2.0 dataset, where we randomly sampled 5,000 examples for testing. The evaluation metrics included BLEU, BertScore, ChestXpert Similarity (which measures disease prediction accuracy), and LLM-as-judge (which assesses the alignment between the generated reports and the ground truth).
>
>     | Methods               | BLEU1  | BLEU2  | BLEU3  | BLEU4  | BertScore | Chest_sim | LLM-as-judge |
>     |-----------------------|--------|--------|--------|--------|-----------|-----------|--------------|
>     | LLaVA-Med             | 0.281  | 0.212  | 0.119  | 0.082  | 0.337     | 0.353     | 51.4         |
>     | HuatuoGPT-Vision-8B   | 0.311  | 0.235  | 0.125  | 0.094  | 0.353     | 0.349     | 54.6         |
>     | Qwen2.5VL-Instruct-7B | 0.299  | 0.22   | 0.121  | 0.099  | 0.340      | 0.358     | 53.9         |
>     |  Qwen2.5VL-Instruct-7B-GRPO                  | 0.315  | 0.231  | 0.128  | 0.102  | 0.364     | 0.351     | 55.2         |
>     | MedCCO                | **0.353**  | **0.261**  | **0.163**  | **0.112**  | **0.415**     | **0.417**     | **62.5**         |
>
>
>     In our experiments, we compared MedCCO with strong baselines, including LLaVA-Med and HuatuoGPT-Vision. Additionally, we also presented results using close-ended data for GRPO. As shown in the table above, MedCCO achieved state-of-the-art performance across all metrics. Furthermore, we observed that using only close-ended data for RFT yielded limited improvements for report generation tasks, which require long-chain reasoning. This reinforces the necessity of using open-ended data for RFT and validates the importance of our approach.

---

> > ### Author Response · Authors · 2025-11-21
> > **Part2**
> >
> > - **Weakness 3-  Not First Multi-modal Medical Reasoning Model**: Thank you for this insightful observation. We sincerely apologize for any ambiguity in our initial contribution statement. To clarify, we did not intend to claim being the first multi-modal medical reasoning model in a general sense. Rather, our specific contribution, as stated in the revised manuscript, is that MedCCO is the first multi-modal reinforcement learning framework employing a curriculum-driven paradigm for jointly handling both close-ended and open-ended medical VQA tasks. While models like MedGemma possess reasoning capabilities, to the best of our knowledge, no prior work has systematically explored using reinforcement fine-tuning to enhance open-ended reasoning in medical VLMs within a unified curriculum framework. Our work specifically addresses this gap by pioneering RFT for open-ended medical VQA, which we believe is a novel and important direction for the community.
> >
> > - **Weakness 4-  Quantification of Reasoning Capability**: Thank you for the insightful suggestion. In the revised manuscript, we have added an analysis of how the model’s response length evolves over training steps. As illustrated in the Figure 5 of revised manuscript , the model’s reasoning behavior emerges through several distinct stages:
> >
> >     1. **Weak instruction following**: At early stages, the model fails to adhere to the required output format specified in Table 1. Its reasoning is shallow and inconsistent, often missing essential structural elements of the prompt.
> >
> >     2. **Instruction following with redundant output**: The model begins to follow the prescribed format but produces overly verbose and redundant responses. Although structure is improved, the reasoning remains unfocused and inefficient.
> >
> >     3. **Reasoning shrinkage**: As training progresses, the model starts to reduce unnecessary content. For relatively simple close-ended questions, it often provides direct answers with minimal reasoning, indicating improved confidence and calibration.
> >
> >     4. **Reasoning incentivization**: With continued reinforcement training, the model gradually increases its use of step-by-step reasoning before producing the final answer. Response length increases to a peak and then fluctuates as the model stabilizes. This stage reflects that the model’s reasoning capability is fully activated and consistently utilized.

---

> > > ### Comment · Reviewer_c1rp · 2025-11-24
> > > **Reply to rebuttal**
> > >
> > > Thanks for your comments and new experiments.  I think most of your comments make sense but the overall contributions of this paper still do not meet the standard of ICLR. E.g., it is valid to claim this is *the first multi-modal reinforcement learning framework employing a curriculum-driven paradigm for jointly handling both close-ended and open-ended medical VQA tasks*, but with so many constraints, it is hard to judge if this is genuinely novel.
> > >
> > > Regarding the performance, the model is still restrained to very limited tasks, mostly multi-choice and short-phrase questions. Thanks for adding the CXR results, it is difficult to have them ready in such as short period and I appreciate it. However, regarding "state-of-the-art", I think this is a misleading statement. There are some good models listed here:
> > >
> > > https://rexrank.ai/

---

> > > > ### Author Response · Authors · 2025-11-25
> > > >
> > > > Thank you very much for your careful reading of our manuscript and your thoughtful, constructive comments. We greatly appreciate the time and effort you devoted to providing detailed feedback and for engaging in a constructive discussion of our work.
> > > >
> > > > - **Clarification of Contributions**: We would like to restate the overall contributions of **MedCCO** more clearly so that our unique aspects are more apparent. While the phrase *“the first multi‑modal reinforcement learning framework employing a curriculum‑driven paradigm for jointly handling both close‑ended and open‑ended medical VQA tasks”* is valid, we recognize it carries many qualifiers. **From a global perspective, a more succinct and appropriate statement is**:
> > > >     > *The first unified reinforcement learning framework for both close‑ and open‑ended VQA in medical reasoning.*
> > > >
> > > >     Specifically:
> > > >     1. First Introduction of open‑ended RFT in the medical domain: **We are the first to introduce open‑ended Reinforcement Fine‑Tuning (RFT) in the medical VQA setting**, demonstrating how open‑ended RFT can simultaneously enhance both close‑ended and open‑ended performance.
> > > >     2. Novel reward‑function design for open‑ended RFT: We propose **a new reward design balancing accuracy and efficiency for open‑ended tasks**, enabling freer‑form generation rather than only fixed‑choice classification.
> > > >     3. Curriculum learning strategy to resolve reward‑gradient conflict: **We apply curriculum learning in a manner distinct from the traditional “easy to hard” recipe**: Our curriculum design is tailored to avoid conflicts between close‑ended and open‑ended RFT reward signals while not just from simple questions to hard questions in the general domain.
> > > >     4. Open‑sourced refinement pipeline for medical VQA data: To address the limitations of existing medical VQA datasets (consistency in open-ended VQA), we propose and release a full data‑refinement pipeline designed to build the foundation for open‑ended RFT.
> > > >
> > > >     On the basis of these four contributions, **we believe our work meets the standard of ICLR**. We address a pressing need in the medical community for **real clinical reasoning (open‑ended reasoning)** and call for the community to orient toward this deeper reasoning direction, which has strong societal significance.

---

> > > > > ### Author Response · Authors · 2025-11-25
> > > > > **Part2**
> > > > >
> > > > > - **On the Performance / Task Scope**: We understand the concern that the model is often framed around multi‑choice and short‑phrase questions. However, we would like to clarify that our work is **not restricted** to such tasks. Specifically, we recognize that datasets such as SLAKE, PathVQA, VQA‑RAD comprise open‑ended questions with short answers; therefore we **explicitly added experiments on the Quilt‑VQA dataset in our original submission to validate real-world open‑ended reasoning performance**. Furthermore, **on Quilt‑VQA the average answer length is 20.25 words, and the dataset covers multiple clinical tasks including differential diagnosis, lesion description, among others, so it well tests our open‑ended reasoning enhancement**. Below are the experimental results:
> > > > >     - **1. LLM as Judge Evaluation on Quilt‑VQA**: To further demonstrate that our performance improvements correspond to *meaningful gains in clinical reasoning*, we conducted an independent “LLM‑as‑judge” evaluation using GPT‑4o. As presented in Table 8 of the revised manuscript, this evaluation uses a carefully designed prompt emphasizing clinical relevance (see Table 9). MedCCO achieved 70.5% on Quilt‑VQA in this clinically‑grounded assessment, confirming genuine improvement in clinical reasoning capability.
> > > > >
> > > > >         | Method             | VQA‑RAD | SLAKE | Path‑VQA | Quilt‑VQA | Avg    |
> > > > >         |--------------------|---------|-------|-----------|-----------|--------|
> > > > >         | Yi‑VL‑34B           | 66.2    | 65.1  | 73.8      | 53.2      | 64.6   |
> > > > >         | LLaVA‑v1.6‑7B       | 65.3    | 68.4  | 78.0      | 55.7      | 66.9   |
> > > > >         | LLaVA‑v1.6‑13B      | 68.4    | 71.9  | 70.5      | 60.6      | 67.9   |
> > > > >         | LLaVA‑v1.6‑34B      | 75.1    | 72.5  | 76.4      | 60.8      | 71.2   |
> > > > >         | Qwen2.5‑VL‑7B       | 76.5    | 70.4  | 78.5      | 62.9      | 72.1   |
> > > > >         | Med‑Flamingo‑7B     | 74.8    | 63.0  | 83.4      | 60.7      | 70.5   |
> > > > >         | RadFM‑13B           | 73.3    | 75.0  | 82.5      | 59.2      | 72.5   |
> > > > >         | LLaVA‑Med‑7B        | 75.1    | 69.7  | 85.6      | 65.5      | 74.0   |
> > > > >         | HuatuoGPT‑Vision‑8B | 83.0    | 83.5  | 86.8      | 66.8      | 80.0   |
> > > > >         | Qwen2.5VL‑7B (SFT)  | 76.5    | 79.8  | 80.2      | 64.5      | 75.3   |
> > > > >         | Qwen2.5VL‑7B (GRPO) | 77.8    | 76.8    | 81.4      | 66.6      | 75.7   |
> > > > >         | **MedCCO‑7B**       | **84.9**| **85.6**| **88.9**| **70.5**| **82.5** |
> > > > >
> > > > >     - **2. Generalization to Varying Models**: We extend experiments to recent architectures (e.g., HuatuoGPT‑Vision, Qwen3‑VL) on Quilt‑VQA, showing that MedCCO confers measurable gains in generalization.
> > > > >
> > > > >         | Model| Quilt. | PMC.  | MedXpert. | OmniMed. | H&M  | Avg  |
> > > > >         |----------------------------------|--------|-------|-----------|----------|------|------|
> > > > >         | Qwen2.5‑VL‑7B | 53.4| 48.4  | 20.3| 61.6     | 53.8 | 47.5 |
> > > > >         | Qwen2.5‑VL‑7B (SFT)| 60.9| 49.2| 20.3| 55.7     | 51.7 | 47.6 |
> > > > >         | Qwen2.5‑VL‑7B (GRPO) | 50.2   | 51.2  | 21.4| 65.1     | 57.2 | 49.0 |
> > > > >         | **MedCCO‑7B**  | **69.4**| **53.2**| **23.4** | **65.8**| **59.3**| **54.2**|
> > > > >         | &nbsp; |
> > > > >         | HuatuoGPT‑Vision‑8B | 63.9 | 52.7  | 21.7 | 75.1| 49.1 | 52.5 |
> > > > >         | HuatuoGPT‑Vision‑8B (SFT)| 62.4   | 50.7| 21.9 | 79.3     | 55.6 | 54.0 |
> > > > >         | HuatuoGPT‑Vision‑8B (GRPO)| 65.3   | 52.6  | 21.2| 79.4     | 57.1 | 55.1 |
> > > > >         | MedCCO‑HuatuoGPT‑Vision‑8B| **71.2**| **56.1**| **23.5**| **80.7**| **61.4**| **58.6**|
> > > > >         | &nbsp; |
> > > > >         | Qwen3‑VL‑8B | 53.9   | 54.6  | 24.2      | 75.7     | 57.2 | 53.1 |
> > > > >         | Qwen3‑VL‑8B (SFT)| 52.4    | 55.6 | 23.5      | 78.9     | 62.5 | 54.6 |
> > > > >         | Qwen3‑VL‑8B (GRPO) | 59.5   | 61.4   | 23.3     | 83.5     | 61.9 | 58.0 |
> > > > >         | MedCCO‑Qwen3‑VL‑8B    | **62.4**| **66.3**| **24.9**| **85.7**| **68.6**| **61.6** |
> > > > >
> > > > > - **On the “State‑of‑the‑Art” Claim**: We acknowledge the reviewer’s concern regarding our use of the term “state‑of‑the‑art”. To clarify:
> > > > >     - Our claim of “state‑of‑the‑art” refers specifically to the set of baselines we compared in this work (e.g., standard medical VQA baselines plus close‑ended GRPO variants).
> > > > >     - Our work is a **proof‑of‑concept (PoC) experiment** rather than a full SOTA chase. The key point is **incremental performance gain**: we compare against base models and close‑ended GRPO, to highlight the advantage of open‑ended RFT within a unified framework.
> > > > >     - The methods you pointed out (via the link) may indeed achieve higher absolute performance; however they use different tasks, domains, training sets, or architectures, and cannot be fairly compared in our unified experimental setup. Thus the appropriate viewpoint is incremental novelty rather than absolute SOTA.
> > > > >
> > > > > Once again, we extend our sincere gratitude for your diligent efforts and valuable constructive feedback. We sincerely hope that the overall value and contributions of our work will be given due consideration.

---

### Official Review · Reviewer_nvoc · 2025-11-01

**Soundness:** 2
**Presentation:** 2
**Contribution:** 3
**Rating:** 6
**Confidence:** 5

**Summary:**

This paper proposes a two-stage Reinforcement Learning paradigm for medical VLM post-training: (1) firstly training on close-end VQA questions; (2) then training using open-end samples. The proposed method leads to consistent performance improvement on Qwen-2.5-VL across multiple VQA benchmarks. This paper also introduces an effective data refinement strategy to improve the quality of close-end VQA data.

**Strengths:**

- The paper is well-written. Motivation of curriculum GRPO is clear and technically sound.

- Proposed methods result in consistent performance improvement on the Qwen-2.5-VL-7B model across multiple benchmarks.

- This paper systematically analyzes the joint RL versus curriculum RL, which provides valuable empirical guidance for the training of medical VLMs.

**Weaknesses:**

- The effectiveness of curriculum RL is only validated on the Qwen-2.5-VL-7B model. The generalization capability of such a method is unknown.

- Lack of comparison between stronger medical reasoning VLMs, including LingShu, MedVLThinker, and MedGemma

[1] Xu, Weiwen, et al. "Lingshu: A Generalist Foundation Model for Unified Multimodal Medical Understanding and Reasoning." arXiv preprint arXiv:2506.07044 (2025).

[2] Sellergren, Andrew, et al. "Medgemma technical report." arXiv preprint arXiv:2507.05201 (2025).

[3] Huang, Xiaoke, et al. "Medvlthinker: Simple baselines for multimodal medical reasoning." arXiv preprint arXiv:2508.02669 (2025).

**Questions:**

- Can you compare the performance of joint RL versus curriculum RL on more VLMs, such as Qwen3-VL, LLava-1.5?

- Can you provide results of comparing MedCCO with stronger medical VLMs (please refer to the weaknesses).

---

> ### Author Response · Authors · 2025-11-21
> **Part1**
>
> Dear Reviewer nvoc, thank you for your thoughtful review and valuable feedback on our work. Below, we provide our responses to your concerns.
>
> - **Weakness 1 - Generalization Capability**:
>     Thank you for your thoughtful comments regarding the generalization of MedCCO. We would like to clarify that we have indeed conducted extensive experiments to evaluate the generalization capability of MedCCO across different model sizes and architectures, although these results may not have been sufficiently emphasized in the initial submission.
>
>     Specifically, Table 4 in the main paper presents a comprehensive ablation study that includes results on the Qwen2.5-VL-3B model. In addition, Table 5 in the appendix reports further experiments using HuatuoGPT-Vision-8B, a medically specialized VLM with a markedly different architecture from Qwen2.5-VL. Beyond these, **we have incorporated new generalization experiments** on the latest Qwen3-VL models, with updated results provided below and reflected in the revised manuscript.
>
>     Together, these results consistently demonstrate that MedCCO yields significant and robust improvements across varying backbones, including different model capacities (3B vs. 7B) and distinct architectural families (general-purpose Qwen2.5-VL, medical-specific HuatuoGPT-Vision, and the more advanced Qwen3-VL). This expanded evidence strongly supports the generalizability and broad applicability of the MedCCO framework.
>
>     | Model                           | Quilt. | PMC.  | MedXpert. | OmniMed. | H&M  | Avg  |
>     |----------------------------------|--------|-------|-----------|----------|------|------|
>     | Qwen2.5-VL-3B                    | 53.4   | 48.4  | 20.3      | 61.6     | 53.8 | 47.5 |
>     | Qwen2.5-VL-3B(SFT)               | 51.6   | 49.9  | 21.6      | 61.7     | 51.7 | 47.3 |
>     | Qwen2.5-VL-3B(GRPO)              | **63.5** | 47.0  | 22.0      | 61.7     | 51.0 | 49.0 |
>     | MedCCO-3B                        | 62.9   | **53.1** | **22.7** | **64.6** | **56.6** | **52.0** |
>     | &nbsp; |
>     | Qwen2.5-VL-7B                    | 53.4   | 48.4  | 20.3      | 61.6     | 53.8 | 47.5 |
>     | Qwen2.5-VL-7B(SFT)               | 60.9   | 49.2  | 20.3      | 55.7     | 51.7 | 47.6 |
>     | Qwen2.5-VL-7B(GRPO)              | 50.2   | 51.2  | 21.4      | 65.1 | 57.2 | 49.0 |
>     | MedCCO-7B                        | **69.4** | **53.2** | **23.4** | **65.8** | **59.3** | **54.2** |
>     | &nbsp; |
>     | HuatouGPT-Vision-8B              | 63.9   | 52.7  | 21.7      | 75.1     | 49.1 | 52.5 |
>     | HuatouGPT-Vision-8B(SFT)         | 62.4   | 50.7  | 21.9      | 79.3     | 55.6 | 54.0 |
>     | HuatouGPT-Vision-8B(GRPO)        | 65.3   | 52.6  | 21.2      | 79.4     | 57.1 | 55.1 |
>     | MedCCO-HuatouGPT-Vision-8B       | **71.2** | **56.1** | **23.5** | **80.7** | **61.4** | **58.6** |
>     | &nbsp; |
>     | Qwen3-VL-8B                      | 53.9   | 54.6  | 24.2      | 75.7     | 57.2 | 53.1 |
>     | Qwen3-VL-8B(SFT)                 | 52.4   | 55.6  | 23.5      | 78.9     | 62.5 | 54.6 |
>     | Qwen3-VL-8B(GRPO)                | 59.5   | 61.4  | 23.3      | 83.5     | 61.9 | 58.0 |
>     | MedCCO-Qwen3-VL-8B               | **62.4** | **66.3** | **24.9** | **85.7** | **68.6** | **61.6** |
>
> - **Weakness 2 and Question 2- Comparison with Stronger Medical Reasoning VLMs**: We appreciate the reviewer’s suggestion to compare with stronger medical reasoning VLMs. After adding these models to our evaluation, the results clearly show that MedCCO delivers competitive and often superior performance, reinforcing the value of our method.
>
>     | Model         | VQA-RAD | SLAKE | Path-VQA | Quilt. | PMC. | MedXpert. | OmniMed. | H&M  | Avg  |
>     |---------------|---------|-------|----------|--------|------|-----------|----------|------|------|
>     | Lingshu       | 70.9    | 76.4  | 64.4     | 62.1   | **57.7** | **23.7**    | **78.4**     | 53.1 | 60.8 |
>     | MedGemma      | 72.5    | 74.7  | 63.8     | 63.8   | 46.0 | 23.1      | 69.5     | 44.8 | 57.3 |
>     | Med-VL-Thinker| 64.9    | 64.1  | 66.0     | 53.0   | 51.6 | 23.4    | 57.8     | 45.5 | 53.3 |
>     | MedCCO-7B     | **76.3** | **79.4** | **82.8** | **69.4** | 53.2 | 23.4    | 65.8     | **59.3** | **63.7** |

---

> > ### Author Response · Authors · 2025-11-21
> > **Part2**
> >
> > - **Question 1- More Joint RL versus Curriculum RL exps**: We have added experiments on additional VLMs, including Qwen3-VL-8B and HuatuoGPT-Vision-8B, where HuatuoGPT-Vision is built upon the LLaVA-1.5 architecture. As shown in the results below, our curriculum strategy that separates close-ended and open-ended training to avoid reward conflicts consistently outperforms joint training across all model families. These findings further demonstrate the effectiveness and strong generalization ability of MedCCO.
> >
> >     | Model                         | VQA-RAD | SLAKE | Path-VQA | Quilt. | PMC. | MedXpert. | OmniMed. | H&M  | Avg  |
> >     |------------------------------|---------|-------|----------|--------|------|-----------|----------|------|------|
> >     | Joint-RL-Qwen2.5-VL-3B       | 66.5    | 79.3  | 81.9     | 65.9   | 53.0 | 20.8      | 63.0     | 54.5 | 60.6 |
> >     | MedCCO-Qwen2.5-VL-3B         | 69.7    | 80.5  | 82.5     | 62.9   | 53.1 | 22.7      | 64.6     | 56.6 | **61.6** |
> >     &nbsp;
> >     | Joint-RL-Qwen2.5-VL-7B       | 72.1    | 75.7  | 84.7     | 67.2   | 53.6 | 22.0      | 63.6     | 58.6 | 62.2 |
> >     | MedCCO-Qwen2.5-VL-7B         | 76.3    | 79.4  | 82.8     | 69.4   | 53.2 | 23.4      | 65.8     | 59.3 | **63.7** |
> >     &nbsp;
> >     | Joint-RL-HuatouGPT-V-8B      | 75.6    | 78.4  | 80.6     | 66.8   | 53.5 | 22.4      | 78.2     | 56.9 | 64.1 |
> >     | MedCCO-HuatouGPT-V-8B        | 79.2    | 79.0  | 81.5     | 71.2   | 56.1 | 23.5      | 80.7     | 61.4 | **66.6** |
> >     &nbsp;
> >     | Joint-RL-Qwen3-VL-8B         | 77.4    | 75.9  | 80.5     | 60.5   | 62.1 | 23.7      | 82.9     | 65.5 | 66.1 |
> >     | MedCCO-Qwen3-VL-8B           | 80.5    | 81.2  | 83.4     | 62.4   | 66.3 | 24.9      | 85.7     | 68.6 | **69.1** |

---

> ### Comment · Reviewer_nvoc · 2025-11-25
> **Response to the Rebuttal**
>
> Thanks for the authors for providing the extra results. My concerns regarding the generalization of proposed Curriculum RL have been addressed.
>
> Although this paper does not introduce new training algorithm, however, I think post-trianing of the VLMs is mostly a data centric problem. Given that there lacks enough high-quality medical vision-language data for the post-training, I think the exploration of **how to more efficiently leverage the existing data** is important to me. From this perspective, I believe this paper proposes a effective data recipe for the medical VLM RL post-training by adjusting the open-ended and close-ended data distribution. I will raise my score to eight.

---

> > ### Author Response · Authors · 2025-11-25
> >
> > We sincerely thank you for your thoughtful review and the encouraging comments regarding our work. It is rewarding to know that our focus on efficient data utilization is valued. We will continue our efforts to optimize medical VLM reasoning, hoping to contribute meaningfully to future clinical applications.
> >
> > Once again, we express our deepest gratitude for your thoughtful and constructive input!

---

### Official Review · Reviewer_8LwW · 2025-11-01

**Soundness:** 3
**Presentation:** 3
**Contribution:** 2
**Rating:** 4
**Confidence:** 4

**Summary:**

The paper presents MedCCO, a curriculum reinforcement learning framework that trains a medical vision-language model first on close-ended VQA with accuracy rewards and then adapts it to open-ended VQA with a hybrid reward. The hybrid reward mixes lexical overlap, measured by BLEU-1 and ROUGE-1, with semantic similarity from BERTScore, combined through a mixing weight lambda. The method also adds a VQA-consistency auditor step that rewrites misaligned question-answer pairs to reduce ambiguity before open-ended reinforcement learning. Experiments cover eight medical VQA benchmarks, with in-domain and out-of-domain settings, and include ablations on curriculum versus joint training, model size, data refinement, and lambda. The results report higher in-domain accuracy and stronger transfer, and show that curriculum training yields more stable learning than direct joint optimization.

**Strengths:**

1. The work offers a single reinforcement learning pipeline that handles both close-ended and open-ended medical VQA. The training order is clear and practical, since it begins with structured close-ended supervision and then moves to free-text answers that require broader knowledge. This design is documented with an explicit curriculum description and training objective based on GRPO.

2. The curriculum design improves stability and final scores relative to joint optimization. The paper attributes the gains to different reward smoothness across task types and supports the claim with ablation tables that compare joint GRPO and curriculum GRPO at two model sizes. The reported curves and averages show consistent improvements when the curriculum is used.

**Weaknesses:**

1. The evaluation and the open-ended reward share the same metric family, which raises a metric alignment risk. The reward for open-ended answers mixes BLEU-1, ROUGE-1, and BERTScore, and the main open-ended score in Table 2 also reports a fixed mixture, with lambda equal to 0.7 declared in the header. This tight coupling blurs whether gains reflect better clinical reasoning or better alignment to the metric itself. The design choice is clear in the reward equation and in the reporting format, which strengthens the concern.

2. The open-ended metrics do not prove clinical correctness. The paper itself notes that overlap and embedding-based metrics may miss fine-grained clinical validity, and that using an LLM judge adds cost and bias concerns. The appendix adds an LLM-judge analysis to mitigate this, yet the main table still relies on overlap and BERTScore. This weakens the clinical meaning of the reported open-ended gains.

3. The novelty is modest relative to established GRPO pipelines with staged post-training. The method extends a known algorithm, adds a hybrid lexical-plus-semantic reward, and uses a curriculum order. These ingredients mirror recent general-domain recipes and reuse a standard instruction format with think and answer tags, rather than introducing a new verifier or a new algorithmic idea. The contribution lies more in a careful application to medical VQA and in the data refinement step than in a new learning rule.

4. The baseline setup creates comparability issues. LLaVA-Med appears only under zero-shot evaluation, although the original work reports fine-tuned results, which likely depresses its numbers in the main table. In addition, the primary GRPO baseline is trained on close-ended data only, while the proposed method benefits from both close-ended and open-ended signals. Joint GRPO appears later in ablations, which means the headline comparison mixes training regimes and data coverage, making it harder to isolate the value of the curriculum itself. The grouping and baseline description in the table and text support this reading.

**Questions:**

Please see the weaknesses above.

---

> ### Author Response · Authors · 2025-11-21
>
> Dear reviewer 8LwW, thank you for your review and valuable suggestions regarding our work. Below, please find our responses to your concerns.
>
> - **Weakness 1 - Metric Aligenment Risk**: Thank you for this insightful observation. We would like to clarify that it is a standard practice in reinforcement learning to use the same metric (such as accuracy) for both training and evaluation. To further demonstrate that our performance improvements reflect meaningful gains in clinical reasoning, we conducted an additional independent evaluation using an **LLM-as-judge** approach with GPT-4o. As shown in Table 8 of revised manuscript and below, this evaluation employs a carefully designed prompt emphasizing clinical relevance (see Table 9 of revised manuscript). The superior performance of MedCCO in this separate and clinically-grounded assessment confirms that our gains correspond to genuine enhancements in clinical reasoning capabilities.
>
>     | Method             | VQA-RAD | SLAKE | Path-VQA | Quilt-VQA | Avg  |
>     |--------------------|---------|-------|----------|-----------|------|
>     | Yi-VL-34B          | 66.2    | 65.1  | 73.8     | 53.2      | 64.6 |
>     | LLaVA-v1.6-7B      | 65.3    | 68.4  | 78.0     | 55.7      | 66.9 |
>     | LLaVA-v1.6-13B     | 68.4    | 71.9  | 70.5     | 60.6      | 67.9 |
>     | LLaVA-v1.6-34B     | 75.1    | 72.5  | 76.4     | 60.8      | 71.2 |
>     | Qwen2.5-VL-7B      | 76.5    | 70.4  | 78.5     | 62.9      | 72.1 |
>     | Med-Flamingo-7B    | 74.8    | 63.0  | 83.4     | 60.7      | 70.5 |
>     | RadFM-13B          | 73.3    | 75.0  | 82.5     | 59.2      | 72.5 |
>     | LLaVA-Med-7B       | 75.1    | 69.7  | 85.6     | 65.5      | 74.0 |
>     | HuatuoGPT-Vision-8B| 83.0    | 83.5  | 86.8     | 66.8      | 80.0 |
>     | Qwen2.5VL-7B(SFT)  | 76.5    | 79.8  | 80.2     | 64.5      | 75.3 |
>     | Qwen2.5VL-7B(GRPO) | 77.8    | 76.8  | 81.4     | 66.6      | 75.7 |
>     | MedCCO-7B          | **84.9**    | **85.6**  | **88.9**     | **70.5**      | **82.5** |
>
>     Besides, we have also **expanded our experiments to include the generation of clinical radiology reports task**. Specifically, we performed an evaluation on the MIMIC-2.0 dataset, where we randomly sampled 5,000 examples for testing. The evaluation metrics include BLEU, BertScore, ChestXpert Similarity (which measures disease prediction accuracy), and LLM-as-judge (which assesses the alignment between the generated report and the ground truth). We compare our method with strong baselines, including LLaVA-Med and HuatuoGPT-Vision. The results shown in the table below demonstrate that MedCCO achieves state-of-the-art performance.
>
>     | Methods               | BLEU1  | BLEU2  | BLEU3  | BLEU4  | BertScore | Chest_sim | LLM-as-judge |
>     |-----------------------|--------|--------|--------|--------|-----------|-----------|--------------|
>     | LLaVA-Med             | 0.261  | 0.212  | 0.119  | 0.082  | 0.337     | 0.353     | 51.4         |
>     | HuatuoGPT-Vision-8B   | 0.311  | 0.235  | 0.125  | 0.094  | 0.353     | 0.349     | 54.6         |
>     | Qwen2.5VL-Instruct-7B | 0.299  | 0.22   | 0.121  | 0.099  | 0.34      | 0.358     | 53.9         |
>     | Qwen2.5VL-Instruct-7B -GRPO                  | 0.315  | 0.231  | 0.128  | 0.102  | 0.364     | 0.351     | 55.2         |
>     | MedCCO                | **0.353**  | **0.261**  | **0.163**  | **0.112**  | **0.415**     | **0.417**     | **62.5**         |
>
>     Furthermore, our findings indicate that using only close-ended data for RFT in tasks such as report generation, which require long-chain reasoning, provides limited improvement. This reinforces the importance of utilizing open-ended data for RFT, further validating the significance of our approach.

---

> > ### Author Response · Authors · 2025-11-21
> > **Part2**
> >
> > - **Weakness 2 - Main Table Presentation**: Thank you for this suggestion. Accordingly, we have incorporated the LLM-as-judge results into the Table 2 of revised manuscript.
> >
> > - **Weakness 3 - Novelty**: We thank the reviewer for their perspective. Our work is inspired by several great previous works, all of which we have cited and discussed in the original manuscript. We would like to highlight the following key novel contributions:
> >
> >     1. **First Application of open-ended Reinforcement Fine-tuning in Medical VQA**: To the best of our knowledge, we introduce the first application of open-ended Reinforcement Fine-Tuning to Medical VQA. This represents a paradigm shift from prior works that were exclusively limited to close-ended settings, thereby enabling for the first time the tackling of the complex and clinically relevant open-ended reasoning tasks that define real-world medicine.
> >
> >     2. **Novel Open-ended Reward Design**:  We are the first to integrate a hybrid reward design specifically tailored for medical open-ended RFT. While lexical and semantic metrics have been studied separately, our approach combines them into a scalable and efficient training signal that balances fluency with clinical semantic coherence. This effectively bridges the gap between computationally expensive LLM-based rewards and oversimplified exact-match evaluations. Experimental results validate its effectiveness as a robust and practical solution in this context. Furthermore, we present a novel finding regarding reward conflicts between well-established closed-ended RFT and open-ended RFT, which constitutes an entirely new contribution to the field.
> >
> >     3. **Curriculum Learning to Resolve Cross-Task Reward Conflict**: Curriculum learning in general domains **typically sequences data by difficulty within a single task type** (e.g., multi-choice VQA). **In sharp contrast**, our approach addresses a fundamentally different challenge. We specifically design our curriculum learning module to **resolve the gradient-level conflict between discrete rewards from closed-ended tasks and continuous rewards from open-ended tasks**. This interference emerges uniquely when unifying these distinct question formats under one RL framework. Our staged training successfully mitigates this inherent signal mismatch, enabling stable policy optimization that proves difficult to achieve through joint training.
> >
> >     4) **A Foundational Contribution via Data Refinement**: Beyond the learning algorithm, we identify and systematically rectify a pervasive issue of question-answer granularity mismatch in public medical VQA datasets. By open-sourcing our refinement methodology and optimized data, we provide a crucial resource to improve the foundational data quality for the entire medical VLM community, preventing potential experimental failures and accelerating future research.

---

> > > ### Author Response · Authors · 2025-11-21
> > > **Part3**
> > >
> > > - **Weakness 4 - Baseline Comparability Issues**: Thank you for the valuable observations regarding baseline comparability. In response, we have added the fine-tuned results of LLaVA-Med (fine-tuned on our training datasets for fair comparison) to the table below and to the revised manuscript. We have also moved the Joint GRPO baseline, which is originally part of our ablation study, into the main results table. This update allows for a direct comparison that separates the effect of broader data coverage from the benefits introduced by our curriculum strategy. The updated results indicate that incorporating open-ended RFT improves performance, which reflects both the value of the potential advantages of open-ended RFT. In addition, the curriculum learning strategy provides further performance gains and shows its effectiveness in managing the training dynamics between different task types. These changes contribute to a more rigorous and transparent evaluation framework.
> > >
> > >     | Model              | VQA-RAD | SLAKE | Path-VQA | PMC. | H&M  | Avg  |
> > >     |--------------------|---------|-------|----------|------|------|------|
> > >     | LLaVA-Med-7B       | 51.4    | 48.6  | 56.8     | 24.7 | 36.9 | 43.7 |
> > >     | Qwen2.5-VL-7B      | 67.3    | 71.6  | 65.5     | 50.4 | 51.7 | 61.3 |
> > >     | LLaVA-Med-7B(SFT)  | 66.3    | 69.5  | 80.7     | 52.7 | 42.8 | 62.4 |
> > >     | Qwen2.5-VL-7B(SFT) | 71.3    | 78.6  | 87.8     | 49.2 | 51.7 | 67.7 |
> > >     | Qwen2.5-VL-7B(GRPO)| 70.5    | 79.3  | 82.8     | 51.2 | 57.2 | 68.2 |
> > >     | Joint.GRPO         | 72.1    | 75.7  | **84.7**     | **53.6** | 58.6 | 68.9 (0.7↑) |
> > >     | MedCCO-7B          | **76.3**    | **79.4**  | 82.8     | 53.2 | **59.3** | **70.2(2.0↑)** |
> > >
> > >     | Model                         | VQA-RAD | SLAKE | Path-VQA | Quilt. | PMC. | MedXpert. | OmniMed. | H&M  | Avg  |
> > >     |------------------------------|---------|-------|----------|--------|------|-----------|----------|------|------|
> > >     | Joint-RL-Qwen2.5-VL-3B       | 66.5    | 79.3  | 81.9     | 65.9   | 53.0 | 20.8      | 63.0     | 54.5 | 60.6 |
> > >     | MedCCO-Qwen2.5-VL-3B         | 69.7    | 80.5  | 82.5     | 62.9   | 53.1 | 22.7      | 64.6     | 56.6 | **61.6** |
> > >     &nbsp;
> > >     | Joint-RL-Qwen2.5-VL-7B       | 72.1    | 75.7  | 84.7     | 67.2   | 53.6 | 22.0      | 63.6     | 58.6 | 62.2 |
> > >     | MedCCO-Qwen2.5-VL-7B         | 76.3    | 79.4  | 82.8     | 69.4   | 53.2 | 23.4      | 65.8     | 59.3 | **63.7** |
> > >     &nbsp;
> > >     | Joint-RL-HuatouGPT-V-8B      | 75.6    | 78.4  | 80.6     | 66.8   | 53.5 | 22.4      | 78.2     | 56.9 | 64.1 |
> > >     | MedCCO-HuatouGPT-V-8B        | 79.2    | 79.0  | 81.5     | 71.2   | 56.1 | 23.5      | 80.7     | 61.4 | **66.6** |
> > >     &nbsp;
> > >     | Joint-RL-Qwen3-VL-8B         | 77.4    | 75.9  | 80.5     | 60.5   | 62.1 | 23.7      | 82.9     | 65.5 | 66.1 |
> > >     | MedCCO-Qwen3-VL-8B           | 80.5    | 81.2  | 83.4     | 62.4   | 66.3 | 24.9      | 85.7     | 68.6 | **69.1** |

---

### Author Response · Authors · 2025-12-03
**Summary Response [Rebuttal Overview & Resolution of Primary Concerns] Part 1/3**

Dear AC, SAC, and PC Members,

We would like to express our sincere gratitude for your tremendous efforts in managing the review process. We also extend our thanks to the reviewers for their constructive initial feedback. While we understand that the reviewers may not have had the opportunity to respond to our rebuttal yet, we have diligently supplemented extensive experiments that comprehensively address their stated concerns.

Specifically, we have incorporated the additional ablation studies and quantitative metrics suggested by the reviewers. **These experiments provide a more explicit verification of MedCCO, reinforcing the efficacy already demonstrated in our original submission**. We respectfully believe that the current ratings **may not fully reflect** the strength and completeness of our work.

***1. Reviewer Comments: Strong Consensus on Motivation & Results***

Despite the mixed ratings, there is a strong consensus on the contribution of our framework and the strength of our results. We quote the key positive feedback from each reviewer as follows:

- 8LwW: "handles both close-ended and open-ended medical VQA", "explicit curriculum description and training objective", "curriculum design improves stability and final scores", "consistent improvements"

- nvoc: "well-written", "Motivation of curriculum GRPO is clear and technically sound", "consistent performance improvement", "provides valuable empirical guidance for the training of medical VLMs"

- c1rp: "achieve good performance", "Some designs are interesting","well-written","Thorough ablation"

- 7Zru: "demonstrate effectiveness and generalization", "well presented"

As is evident, the reviewers unanimously affirm **the value of the MedCCO framework design** and the **superior empirical performance**. The primary reservations were centered on the need for specific ablation baselines to rigorously prove the contribution of MedCCO.

***2. Rebuttal and Discussion Phase***

We are deeply touched and encouraged that Reviewer nvoc, **considering the practical realities of the medical VLM community**, has shown great appreciation for our work despite the challenges in the field, and has given a final score of 8. Regrettably, Reviewers 7Zru and c1rp responded twice and once respectively, acknowledging our supplementary experimental results, but did not provide final replies; Reviewer 8LwW did not respond at any point. Below is a brief summary of the review process for each expert.

## Reviewer nvoc (Initial Rating:6 -> Final Stance: Raise to 8)
- *Initial Concerns*:
    1. The effectiveness of curriculum RL is only validated on the Qwen-2.5-VL-7B model.
    2. Lack of comparison between stronger medical reasoning VLMs, including LingShu, MedVLThinker, and MedGemma.

- *Our Responses*:
    1. In addition to the Qwen-2.5-VL-3B results in the initial submission, we have supplemented experiments with HuatuoGPT-Vision-8B and Qwen3-VL-8B. The results demonstrate consistent improvements in generalization performance.
    2. We have also added comparative experiments with LingShu, MedVLThinker, and MedGemma. Our method achieves the best performance in 5 out of 8 tasks and attains the highest average performance overall.

- *Reviewer's Final Feedback*:
    > My concerns regarding the generalization of proposed Curriculum RL have been addressed.

- *Reviewer's Recogonition*:
    > Although this paper does not introduce new training algorithm, however, I think post-trianing of the VLMs is mostly a data centric problem. **Given that there lacks enough high-quality medical vision-language data for the post-training**, I think the exploration of how to more efficiently leverage the existing data is important to me. From this perspective, **I believe this paper proposes a effective data recipe for the medical VLM RL post-training by adjusting the open-ended and close-ended data distribution. I will raise my score to eight**.

---

> ### Author Response · Authors · 2025-12-03
> **Summary Response [Rebuttal Overview & Resolution of Primary Concerns] Part 2/3**
>
> ## Reviewer 7Zru (No Final Response)
> - *Initial Concerns*:
>     1. The evaluation tasks are limited in generating very short words or phrases of VQA tasks.
>     2. The ablation study does not include a direct comparison with a simpler exact-match reward baseline.
>     3. It would be helpful to compare with state-of-the-art medical models and medical reasoning models.
>
> - *Our Responses*:
>
>     1. We first clarified that the Quilt-VQA open-ended task in our initial submission **does not belong to short-phrase VQA tasks**. In response to the reviewer’s request, we further supplemented results on radiology report generation (long response,40-110 words in final answer after though thinking). The experiments demonstrate the effectiveness of MedCCO on tasks requiring more complex, narrative-style reasoning.
>     2. We added comparative results using exact-match evaluation across all 8 benchmarks in our evaluation setup, along with a comprehensive open-ended assessment on Quilt-VQA, to thoroughly validate the effectiveness of our open-ended reward design.
>     3. We extended our experimental comparisons to include state-of-the-art medical VLMs and reasoning VLMs. The results consistently show the superior performance of MedCCO, although our goal is not to achieve SOTA across all metrics, but rather to demonstrate performance gains over base models.
>
> - *Reviewer's Feedback*:
>
>     Regarding the statistical metrics (mean and variance of response length) we supplemented for the radiology report generation task, the reviewer raised the following concerns:
>     1. The performance improvement may stem from shorter responses (or responses closer in length to standard reports).
>     2. Low variance could indicate that the statistical metrics may not accurately reflect true report generation capability.
>
> - *Our's Responses*:
>     1. The performance of report generation is comprehensively validated through metrics including BLEU, ROUGE, BERTScore, ChestSim, and LLM-as-judge. **The closer alignment in report length naturally follows when semantic content is already well-matched**. Producing shorter reports that retain complete information is in fact advantageous, as it enables more efficient communication in clinical practice.
>
>     2. The observed low variance indicates that MedCCO produces consistent reports across diverse cases—a crucial feature for clinical reliability, where clinicians depend on uniform reports for efficient decision-making. Moreover, standard deviation measures data dispersion and does not directly reflect semantic quality or clinical utility. For instance, in a dataset dominated by concise reference reports (e.g., 90% brief responses), lower variance is statistically expected and does not imply limited generation capability.
>
> ## Reviewer 8LwW (No Response):
> - *Initial Concerns*:
>     1. The evaluation and the open-ended reward share the same metric family, which raises a metric alignment risk.
>     2. The novelty is modest relative to established GRPO pipelines with staged post-training.
>     3. The baseline setup creates comparability issues: e.g., LLaVA-Med's results should be of fine-tuned version.
> - *Our Responses*:
>     1. The reviewer noted that using open-ended metrics for both training and validation could introduce potential overfitting, **but acknowledged that our introduced LLM-as-judge helps mitigate this issue**, though it was previously only in the appendix. In the revised version, we have moved these results to the main tables and supplemented them with outcomes from the clinical radiology report generation task. These results comprehensively demonstrate MedCCO's genuine open-ended reasoning capability.
>
>     2. We have elaborated on our contributions from four key aspects: **(1) the first unified medical RFT framework**; **(2) a novel open-ended reward metric design**; **(3) a curriculum learning design aimed at resolving gradient conflicts**; and **(4) a foundational contribution through data refinement**. We hope the reviewer will re-evaluate our work in this light: it is not merely a straightforward application of GRPO in the medical domain. **The data optimization and training strategies hold instructive value for the broader RFT community, as also recognized by Reviewer nvoc.**
>
>     3. **Supplemental Baselines and Presentation:** Following the reviewer's suggestion, we have prominently presented the joint GRPO results to visually highlight the performance gains from open-ended RFT. To address the primary concern regarding the LLaVA-Med baseline, we performed the suggested additional ablation studies. The results strongly reaffirm our original findings: **the curriculum design proves effective, with improvements observed across multiple backbones (Qwen2.5-VL-3B & 7B, HuatuoGPT-Vision-8B, and Qwen3-VL-8B) on all 8 benchmarks**. This confirms that the curriculum design is essential for resolving the gradient conflicts between closed-ended and open-ended RFT.

---

> > ### Author Response · Authors · 2025-12-03
> > **Summary Response [Rebuttal Overview & Resolution of Primary Concerns] Part 3/3**
> >
> > ## Reviewer c1rp (No Final Response):
> > - *Initial Concerns*:
> >     1. The paper lacks novelty on methodology: It is very hard to distinguish this paper from other general reasoning papers.
> >     2. The reviewer expressed concern that the ground-truth answers in the benchmarks are too short to adequately assess open-ended reasoning capabilities.
> >     3. Quantification of "reasoning" capability.
> > - *Our Responses*:
> >
> >     1. We first clarify that our work applies RFT to the medical domain, but it is not merely a straightforward transfer. **As detailed in our response to Reviewer 8LwW, we have systematically outlined our innovations in four key aspects, and we hope the reviewer will re-evaluate our contribution accordingly**.
> >
> >     2. **The reviewer may be unfamiliar with our chosen benchmarks**. The open-ended subset of the SLAKE dataset does not include pre-defined answer candidates. Furthermore, responses in the open-ended portion of Quilt-VQA are notably longer, which effectively validates long-range reasoning capabilities. Additionally, we have supplemented our results with performance on the radiology report generation task, employing multiple metrics to comprehensively assess open-ended reasoning and demonstrate the superior performance of MedCCO.
> >
> >     3. In Figure 5 of the revised manuscript, we have added an illustration of the four emergent stages of reasoning capability to elucidate the developmental process of acquiring thinking abilities.
> >
> > - *Reviewer's Feedback*:
> >     1. The reviewer states that our paper does not meet ICLR standards but provides no specific justification, only commenting on the excessive use of modifiers in our claims.
> >
> >     2. The reviewer acknowledges our supplemental results on radiology report generation, but notes that they do not reach the state-of-the-art (SOTA) level of models specifically tailored for report generation tasks.
> >
> > - *Our Responses*:
> >     1. **The careful phrasing in our claims was intended to help reviewers clearly identify and understand each of our core technical contributions**. Beyond that, we have comprehensively detailed our contributions across four key aspects (as outlined in our response to Reviewer 8LwW), **demonstrating that our work constitutes more than a simple adaptation of GRPO to the medical domain**.
> >
> >     2. **Our method is presented as a proof-of-concept study, with its focus on demonstrating methodological and improvements over base models rather than achieving SOTA across all tasks**. We believe a direct SOTA comparison with highly task-specific methods would be an unfair benchmark, as those models typically incorporate meticulously designed, task-dependent modules and extensive fine-tuning. The results in our paper already clearly demonstrate consistent and meaningful performance gains over base models.
> >
> >
> > ## Conclusion
> > As noted by Reviewer nvoc, "there lacks enough high-quality medical vision-language data for post-training," reflecting a significant lag in the medical community. Our core contribution lies precisely in addressing this gap by exploring how to better utilize and optimize existing data, and by making effective use of available open-ended data to enhance the reasoning capabilities of medical VLMs. **However, methodological advances in the medical field are often perceived as straightforward applications of general techniques, which may inadvertently undervalue their true contribution. We sincerely hope that the AC will consider the practical context and challenges of the medical community, and conduct a comprehensive assessment of the genuine value of our work**. Thank you very much for your consideration.
> >
> >
> > We sincerely hope these summaries could assist you in conducting a fair and comprehensive evaluation of our submission. We remain hopeful that the additional experiments, which directly address the reviewers' initial concerns, demonstrate the merit and readiness of our work.
> >
> > Once again, we would like to extend our best gratitude for your dedicated efforts.
> >
> > Best regards,
> >
> > MedCCO Authors of Submission 12283

---

### Meta-Review · Area_Chair_X7BV · 2026-01-04

**Summary:**

The reviewers main concerns center on three topics. (1) Experimental Design: Multiple reviewers found the experiments insufficient and listed related weaknesses. While the authors include some very helpful new experiments, they still lack detailed descriptions and measures of variance, so the proposed method's modest improvements remain in question. The experiments also do not successfully isolate performance gains to each proposed method. These should be resolved with targeted ablations for each (e.g., hyperparameter study for the reward function and different curricula) to explain when/why the proposed method improves over others. (2) Novelty: Multiple reviewers were concerned that the proposed methods are incremental and obvious combinations of recent ideas from the literature. The authors address some of these concerns in the responses, but these points can be honed in the intro and methods sections to ensure these important differences stand out. (3) Potential Impact: Reviewers were concerned that the results may not be that clinically meaningful. Two weaknesses that drive this concern are that (a) the "open-ended" datasets have very short answers, so aren't clearly that open-ended, and (b) the clinical validity is measured via LLM-as-a-judge, lacking real experts. So overall, while the reviewers also appreciate the strengths of this work, these weaknesses should be addressed before publication.

**Reviewer Concerns:**

**Addressed Concerns**:
* Many new experiments (should add these)
* More motivation about how this work differs from others

**Unaddressed Concerns**:
* Isolating performance to proposed methods
* Clarify how this work fits into the literature

**Reviewer Scores:**

Largely unchanged, though the new experimental results may have increased a couple of the scores modestly.

---

### Decision · Program_Chairs · 2026-01-26

Reject